# An along-track biogeochemical Argo modelling framework, a case study of model improvements for the Nordic Seas

Veli Çağlar Yumruktepe[1], Erik Askov Mousing[2], Jerry Tjiputra[3], and Annette Samuelsen[1]

[1]Nansen Environmental and Remote Sensing Centre and Bjerknes Centre for Climate Research, Jahnebakken 3, N-5007, Bergen, Norway
[2]Institute of Marine Research, Bjerknes Centre for Climate Research, Nordnesgaten 33, NO-5005, Bergen, Norway
[3]NORCE Norwegian Research Centre, Bjerknes Centre for Climate Research, Jahnebakken 5, N-5007, Bergen, Norway

**Correspondence:** Veli Çağlar Yumruktepe (caglar.yumruktepe@nersc.no)

**Abstract.** We present a framework that links in situ observations from the biogeochemical-Argo (BGC-Argo) array to biogeochemical models. The framework allows a minimized technical effort to construct a Lagrangian type 1D modelling experiment along BGC-Argo tracks. We utilize the Argo data in two ways; (1) drive the model physics, (2) evaluate the model biogeochemistry. BGC-Argo physics data is used to nudge the model physics closer to observations to reduce the errors in biogeochemistry stemming from physics errors. This allows us to target model biogeochemistry and by using the Argo biogeochemical dataset, we identify potential sources of model errors, introduce changes to model formulation, and validate model configurations. We present experiments for the Nordic Seas and showcase how we identify potential BGC-Argo buoys to model, prepare forcing, design experiments and approach model improvement and validation. We used ECOSMO II(CHL) model as the biogechemical component and focused on chlorophyll *a*. The experiments revealed that ECOSMO II(CHL) required improvements during low-light conditions, as the comparison to BGC-Argo reveals that ECOSMO II(CHL) simulates a late spring bloom and does not represent the deep chlorophyll maximum formation in summer periods. We modified the productivity and chlorophyll *a* relationship and statistically documented decreased bias and error in the revised model using BGC-Argo data. Our results reveal that nudging the model T and S closer to BGC-Argo data reduces errors in biogeochemistry, and we suggest a relaxation time-period of 1 - 10 days. The BGC-Argo data coverage is ever growing and the framework is a valuable asset for improving models in 1D-model efficiently and transfer the configurations to 3D-model with a wide range of focus from operational, regional/global and climate scale.

## 1 Introduction

Marine biogeochemical models are used to understand and quantify the physical, chemical, and biological interactions and how they respond or feedback to climate variability. Ocean biogeochemistry is complex with many poorly known processes, and it is therefore necessary to simplify the ecosystem functions to construct modelling frameworks that represent the environment they are dedicated to in a cost-efficient way. In addition to observational datasets, we require efficient tools that can maximize these datasets' benefits for model construction, tuning and evaluation. In this study, we showcase how biogeochemical-Argo buoys can be used for improving model formulation, parameterisation and performance in a biogeochemical model.

Biogeochemical-Argo (BGC-Argo) is a network of free-drifting, battery-powered profiling floats measuring temperature, salinity, as well as six core variables down to a depth of 2,000 m: oxygen, nitrate, pH, chlorophyll *a*, suspended particles, and downwelling irradiance. In the context of Global Ocean Observing System, the BGC-Argo network supports three main themes: climate, marine ecosystem health, and operational services. It has been used to estimate and assess the net community and export production, the air-sea gas exchange, oxygen minimum zone variability, and biophysical interactions (Claustre et al., 2020, and references therein). In the modelling community, BGC-Argo datasets have been used with data assimilation for state correction (Cossarini et al., 2019), model optimisation and evaluation (Verdy and Mazloff, 2017; Damien et al., 2018; Salon et al., 2019; Wang et al., 2020) as well as model formulation improvement (Terzić et al., 2019). These recent examples demonstrate the synergy between BGC-Argo with biogeochemical models and that the use of BGC-Argo in the modelling community is gaining momentum. Furthermore, the added value of these profiling floats to modelling frameworks will likely increase with increasing BGC-Argo coverage (Voosen, 2020). Therefore, introducing frameworks (including the one presented in this study) utilising these datasets will benefit the modelling community as both the regional and temporal BGC-Argo coverage increases.

During its two decades of operation, the BGC-Argo array has challenged the capacity of historical in situ sampling which is biased towards coastal areas, the Northern Hemisphere and seasons of easier sampling conditions, especially in the polar regions (Riser et al., 2016). BGC-Argo covers open-ocean regions extensively and sample equally throughout the seasons (see Section 2.1.1 for the sampling comparison in the high latitude North Atlantic). For 1D models, which are preferably configured at the time-series sites, the overhead for in situ sampling exceedingly limits the temporal resolution, leading to under-sampling which limits understanding the ocean dynamics and model development and assessment. While satellite images provide extensive regional coverage (hindered by cloud coverage, especially at the high latitudes), they are limited to surface measurements and alone can not constrain some of the vital parameters for model optimisation and validation (Tjiputra et al., 2007; Gharamti et al., 2017; Wang et al., 2020).

In this study, we focus on using BGC-Argo as an additional observational data source to in situ sampling and remote sensing, and how to take advantage of two important aspects of the BGC-Argo dataset: (1) its regional and temporal coverage, (2) combined availability of high-resolution physical and biogeochemical data. Our main objective is to establish the framework and showcase its capacity as a tool for model development and assessment. The framework will allow the modeller to construct a Lagrangian type experiment along a BGC-Argo track in order to visually and objectively assesses the model performance and subsequently advance its dynamics and optimise its parameters. Even though one of the ultimate aims of using this framework for a modelling study is the assessment of the observed biogeochemistry, our primary aim is to present the details of the framework. Therefore a full assessment of the observed biogeochemical variables is outside the scope of this study. Here, we present how BGC-Argo physical data can enhance the realism of model physics, thereby allowing the evaluation and improvement of the modeled biogeochemistry. Specifically, we show how its high-resolution vertical and temporal chlorophyll *a* sampling can be used to advance model formulation, and objectively assess the model parameters. The ultimate goal is to establish a 1D modelling framework towards improving regional and global models.

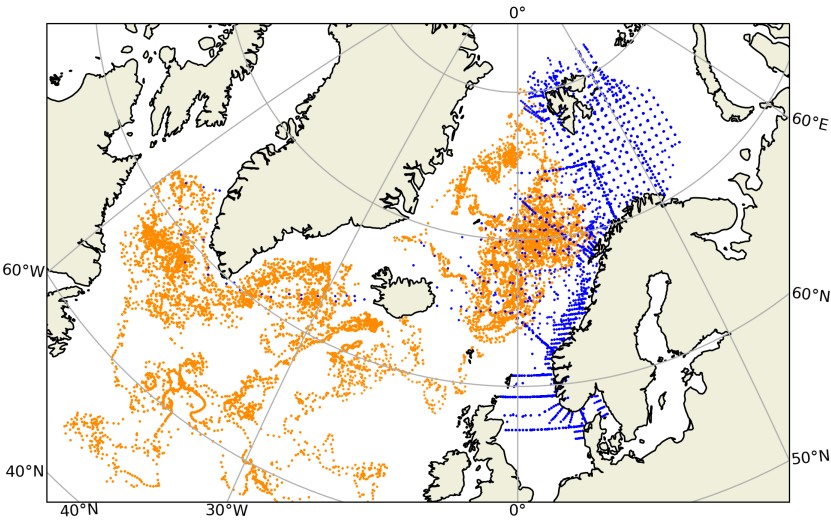

**Figure 1.** Spatial distribution of the BGC-Argo profiles (orange) and in situ sample (blue) locations in the northern North Atlantic. The two observing platforms have too few spatio-temporal matches to perform a meaningful combined statistical analysis. For this comparison, in situ samples from 2010 - 2017 have been used.

## 2 Materials and Methods

### 2.1 Observation datasets

#### 2.1.1 Biogeochemical-Argo

BGC-Argo data were downloaded from the Copernicus Marine Services web portal (marine.copernicus.eu) under the Global Ocean – Delayed Mode Biogeochemical product category (INSITU GLO BGC DISCRETE MY OBSERVATIONS 013 046; https://doi.org/10.17882/86207) as separate NetCDF files for each BGC-Argo buoy. A regional filter was applied to this dataset, so that only BGC-Argo buoys that were located in the North Atlantic and above $50^{o}$N latitude at any time during its course were selected (Fig.1). We only selected buoys that include the CPHL_ADJUSTED variable (from now on chlorophyll *a* unless stated otherwise), indicating that a correction has been applied to the chlorophyll *a* data. A visual inspection was performed, and only those BGC-Argo buoys with chlorophyll *a* profiles that can represent a continuous temporal and depth coverage were selected. A total of 53 BGC-Argo buoys were selected for statistical validation of their chlorophyll *a* (see Section 3.1). Throughout this selection process, BGC-Argo chlorophyll *a*, salinity and temperature data with 1, 5 and 8 quality control flags were used. They represent good, adjusted, and interpolated data respectively. A final inspection was made and 8 buoys (see Supplementary materials for the buoys used) were selected for the along-track simulations considering the most feature-rich buoys with multiple year continuous chlorophyll *a* coverage. We present the results from the buoy 6902547 in the main text.

### 2.1.2 Satellite and in situ chlorophyll *a*

Ocean Colour Climate Change Initiative (OC CCI v5.0) daily L3 chlorophyll *a* and kd$_{490}$ data (Sathyendranath et al., 2019, 2021) were read from the thredds server (https://rsg.pml.ac.uk/thredds/catalog/cci/v5.0-release/geographic/daily/catalog.html) as sub-set datasets around the coordinates of either BGC-Argo buoy profiles or surface in situ samples. Chlorophyll *a* samples collected by the Institute of Marine Research (2018) were used for the statistical evaluation of the BGC-Argo chlorophyll *a* data (http://www.imr.no/forskning/forskningsdata/infrastruktur/viewdataset.html?dataset_id=104).

### 2.2 Biogeochemical Argo, satellite and in situ data co-locating procedure and analysis

We performed a cross-validation of the BGC-Argo chlorophyll *a* in our study region ($> 50^o$N) against in situ samples using satellite data as reference for both BGC-Argo and in situ chlorophyll *a*. We performed the statistical analysis using the satellite data because the number of co-locations between the BGC-Argo and in situ samples were not enough for a representative statistical analysis. The satellite chlorophyll *a* and kd$_{490}$ data were retrieved from the thredds server and the satellite data were averaged within 2 km radius around the BGC-Argo and in situ profile coordinates. Both BGC-Argo and in situ chlorophyll *a* profiles were averaged within 1/kd$_{490}$ (m) depth if kd$_{490}$ data was available within the 2 km radius. If kd$_{490}$ data was missing, profiles were averaged within 10 meters depth. For the statistical analysis, bias, root mean square error (rmse), correlation (corr) and normalized standard deviations (nstd) were calculated for the co-located data following:

$$bias = \left( \sum (M - O) \right) / \sum O, \tag{1}$$

$$rmse = \sqrt{\sum (M - O)^2 / N}, \tag{2}$$

$$corr = \frac{\sum_{i=1}^{N} (M_i - \overline{M})(O_i - \overline{O})}{\sqrt{\sum_{i=1}^{N} (M_i - \overline{M})^2 \sum_{i=1}^{N} (O_i - \overline{O})^2}}, \tag{3}$$

$$nstd = \frac{\sqrt{\left( \sum_{i=1}^{N} (M_i - \overline{M})^2 \right)}}{\sqrt{\left( \sum_{i=1}^{N} (O_i - \overline{O})^2 \right)}} \tag{4}$$

where M = estimated, O = observed, N = number of data points and i = individual sample.

### 2.3 Model description

Physical processes in the water column were simulated by the 1D General Ocean Turbulence Model (GOTM; Burchard et al., 1999) which simulates vertical turbulent fluxes of momentum, heat and dissolved and particulate matter. All experiments described in the manuscript used 190 vertical layers (2000 meters deep) of varying thicknesses with thin layers near the surface and the bottom of the water column. A 1-hour resolution atmospheric forcing was applied. We applied the GOTM model default turbulence closure ($2^{nd}$ order) method with k-epsilon style turbulence kinetic energy equation. Physics was simulated along the BGC-Argo track with the assumption that the lateral interactions in the water column were minimized as the buoys travelled

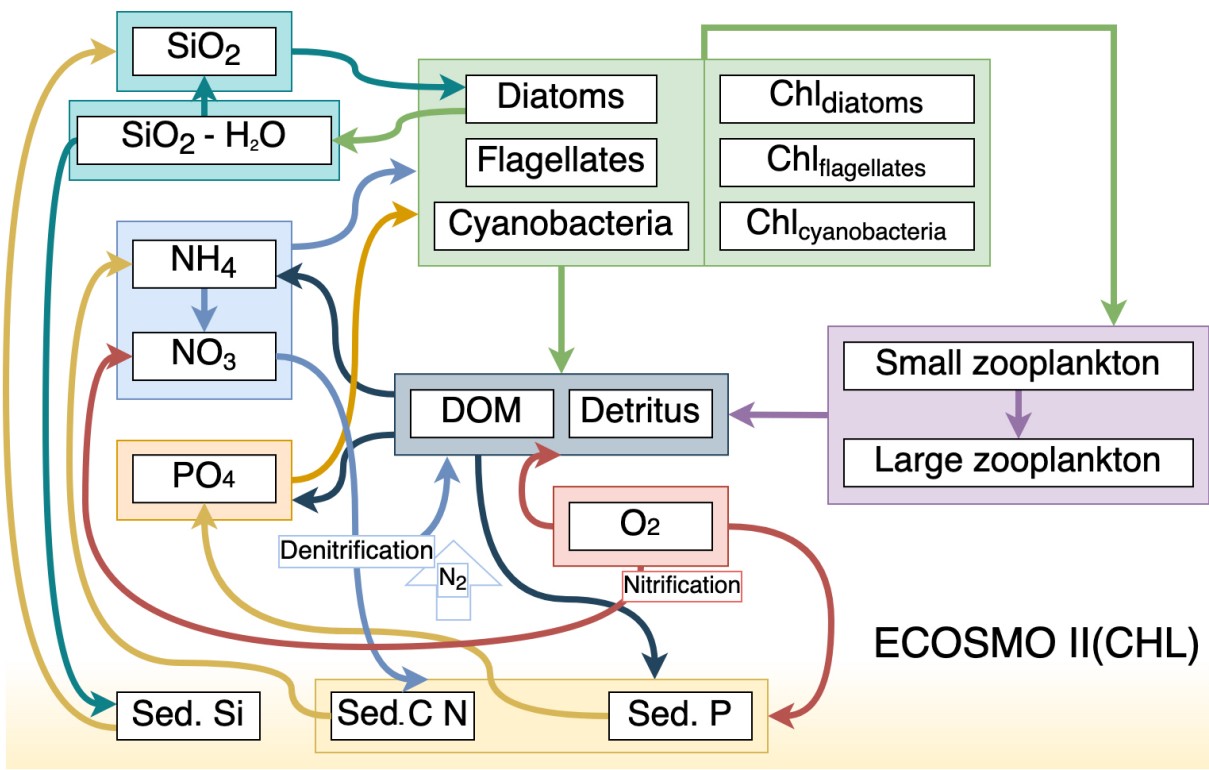

**Figure 2.** Schematic diagram (after Yumruktepe et al., 2022b) of biochemical interactions in ECOSMO II(CHL) (DOM: dissolved organic matter; Chl- prefixes stand for phytoplankton type specific chlorophyll a content; Sed. denote sediment pool with silicate, phosphorus and nitrate content.)

together with the water mass they were located in. However, certain lateral interactions were included through relaxation to prescribed datasets (see Section 2.4.1).

We used ECOSMO II(CHL) (Y2022; Yumruktepe et al., 2022b) to simulate the biogeochemical processes in the water column. ECOSMO II(CHL) is an intermediate-complexity lower trophic level biogeochemical model resolving four inorganic nutrients (nitrate, ammonium, phosphate and silicate) utilized by three types of phytoplankton (diatoms, flagellates and cyanobacteria). In this study, cyanobacteria were turned off since they were parameterized to grow in the Baltic Sea (DS2013; Daewel and Schrum, 2013). North of $50^o$N in the North Atlantic, cyanobacteria is not a significant component of the phytoplankton community. Two types of zooplankton (micro- and meso-size classes) are parameterized as herbivores and omnivores respectively. Dissolved (DOM) and particulate (detritus) organic matter are included in the model. The model uses the molar Redfield ratio between C:N:Si:P components (106 : 6.625 : 6.625 : 1), and discrete nutrients are tracked both in the water column and in a single sediment layer. A complete description of ECOSMO II is given in Daewel and Schrum (2013). ECOSMO II(CHL) version Y2022 includes chlorophyll *a* as an explicit state variable for each phytoplankton functional type. Element flow between ECOSMO II(CHL) variables are given in Figure 2.

ECOSMO II(CHL) formulated in Y2022 represents the current (December 2022) operational marine biogeochemical model for the Nordic Seas and the Arctic Ocean (ARC MFC – Arctic Marine Forecasting Centre) under the umbrella of The Copernicus Marine Services (marine.copernicus.eu; https://doi.org/10.48670/moi-00003). This model's formulation and its parameterisation was used as the reference model for the experiments conducted in this study and is referred to as 'REF'. While Y2022 presents the upgrades to DS2013 version with the addition of an explicit chlorophyll $a$ variable for each phytoplankton functional type, Yumruktepe et al. (2022b) model evaluation reveal that further refinement of chlorophyll $a$ is needed to improve its dynamical response to varying light conditions. We address this issue while presenting the use case of along-track BGC-Argo modelling framework. Thus, we have introduced changes to the light-limitation on phytoplankton growth formulation and to various parameters and these changes will be validated using the BGC-Argo data.

ECOSMO II(CHL) formulates the biological interaction for phytoplankton types $R_{Phy_j}$ and chlorophyll $a$ $R_{Chl_j}$ for $P_1$ and $P_2$ (diatoms and flagellates respectively) as the following:

$$R_{Phy_j} = \sigma_j \phi_{P_j} C_{P_j} - \sum_{i=1}^{2} G_i P_j C_{z_i} - m_{p_j} C_{P_j}, \tag{5}$$

$$R_{Chl_j} = \rho_{chl_j} \sigma_j \phi_{P_j} C_{P_j} - \sum_{i=1}^{2} G_i P_j C_{z_i} \frac{Chl_{P_j}}{C_{P_j}} - m_{p_j} Chl_{P_j} \tag{6}$$

where

$$\rho_{chl_j} = \frac{\theta_{P_j}^{max} \phi_{P_j} C_{P_j}}{\alpha_{P_j} I(x,y,z,t) Chl_j}, \tag{7}$$

$$\phi_{P_j} = min(\alpha_j(I), \beta_N, \beta_P, \beta_{Si}), \tag{8}$$

$$\alpha_j(I) = tanh(\varphi_j I(x,y,z,t)), \tag{9}$$

$$\beta_N = \beta_{NH_4} + \beta_{NO_3}, \tag{10}$$

$$\beta_{NH_4} = NH_4/(NH_4 + r_{NH_4}), \tag{11}$$

$$\beta_{NO_3} = (NO_3/(NO_3 + r_{NO_3}))exp(-\gamma NH_4), \tag{12}$$

$$\beta_{PO_4} = PO_4/(PO_4 + r_{PO_4}), \tag{13}$$

$$\beta_{Si} = Si/(Si + r_{Si}), \tag{14}$$

$$G_i P_j = \sigma_{i,P_j} \frac{a_{i,P_j} C_{P_j}}{r_i + F_i}, \tag{15}$$

$$F_i = \sum_{j=1}^{2} a_{i,P_j} C_{P_j} \tag{16}$$

with $j = 1,2$ denote the specific phytoplankton types and $i = 1,2$ the specific zooplankton types. $P$ (phytoplankton) and $Z$ (zooplankton) concentrations in [mg m$^{-3}$] are represented by $C$, while $Chl$ denote chlorophyll $a$ concentration in mg m$^{-3}$. Silicate is not included in flagellate equations. The parameter definitions and units are provided in Table 1. Photosynthetically

**Table 1.** Symbol definitions of Eqs. 5 - 18

| symbol | description | unit |
|---|---|---|
| $C_{P_j}$ | phytoplankton biomass | mgC m$^{-3}$ |
| $C_{z_i}$ | zooplankton biomass | mgC m$^{-3}$ |
| $Chl_{P_j}$ | chlorophyll $a$ concentration | mgChl m$^{-3}$ |
| $R_{Phy_j}$ | phytoplankton biomass sources/sinks | mgC m$^{-3}$ d$^{-1}$ |
| $R_{Chl_j}$ | chlorophyll $a$ biomass sources/sinks | mgChl m$^{-3}$ d$^{-1}$ |
| $\sigma_j$ | phytoplankton maximum growth rate | d$^{-1}$ |
| $\phi_{P_j}$ | growth limitation | |
| $m_{p_j}$ | phytoplankton mortality rate | d$^{-1}$ |
| $\theta_{P_j}^{max}$ | maximum Chl:C ratio | mgChl mgC$^{-1}$ |
| $\alpha_{P_j}$ | initial slope of P-I curve | mgC m$^2$ (mgChl d W)$^{-1}$ |
| $I(x,y,z,t)$ | photosynthetically active radiation (PAR) | W m$^{-2}$ |
| $\varphi_j$ | photosynthesis efficiency parameter | m$^2$ W$^{-1}$ |
| $r_{NH_4,NO_3,PO_4,Si}$ | nutrient-specific half saturation constant | mmol(N,P,Si) m$^{-3}$ |
| $\gamma$ | NH$_4$ inhibition | m$^3$ molN$^{-1}$ |
| $\sigma_{i,P_j}$ | zooplankton specific grazing rate | d$^{-1}$ |
| $a_{i,P_j}$ | zooplankton food preference | |
| $r_i$ | half saturation constant for grazing | mgC m$^{-3}$ |
| $k_w$ | light attenuation due to water constant | m$^{-1}$ |
| $k_{Chl}$ | light attenuation due to chlorophyll $a$ concentration constant | m$^2$ mgChl$^{-1}$ |

$j$ = 1,2 denote the specific phytoplankton types and $i$ = 1,2 the specific zooplankton types. Please refer to Yumruktepe et al. (2022b) and Daewel and Schrum (2013) for the parameter values that are not given Table 2.

active radiation (PAR) is defined as $I(x,y,z,t)$ and is formulated as:

$$I(z) = 0.42 * I_s * exp(-k_w z - k_{chl} \int\limits_z^0 Chl_{P_j} \partial z) \tag{17}$$

While the chlorophyll $a$ was introduced as an explicit variable in Y2022, its effect on phytoplankton growth was only indirectly included through Eq.17 as a self-shading parameter on light attenuation. Variability under varying light conditions was allowed by defining C:Chl ratio as a function of light (Eq.7). In this study, we expand on the Y2022 approach and include chlorophyll $a$ to have a direct effect on phytoplankton growth, e.g. low-light conditions trigger higher production of chlorophyll $a$ (was present in Y2022) and will increase production (introduced in this study) due to higher chlorophyll $a$ concentration resembling increased used of light energy. This was applied in 'EXP' experiments by modifying $\alpha_j(I)$ in Eq. 9, the light limitation on growth, with that following the formulation of Evans and Parslow (1985) and parameterisation of Bagniewski

**Table 2.** Modified ECOSMO II(CHL) parameters between REF and EXP configurations

| Model parameters | REF | EXP |
|---|---|---|
| max growth rate for diatoms ($\sigma_j$) (d$^{-1}$) | 1.75 | 1.15 |
| max growth rate for flagellates ($\sigma_j$) (d$^{-1}$) | 1.45 | 1.0 |
| max. grazing rate of mesozooplankton on phytoplankton ($\sigma_{i,P_j}$) (d$^{-1}$) | 1.2 | 0.8 |
| max. grazing rate of microzooplankton on phytoplankton ($\sigma_{i,P_j}$) (d$^{-1}$) | 1.5 | 1.0 |
| max. grazing rate of mesozooplankton on microzooplankton ($\sigma_{i,P_j}$) (d$^{-1}$) | 0.75 | 0.5 |
| Zooplankton half-saturation constant for grazing ($r_i$) (mmolN m$^{-3}$) | 0.5 | 0.3 |
| Mesozooplankton mortality rate ($m_{Z_i}$; Daewel and Schrum (2013) ) (d$^{-1}$) | 0.2 | 0.08 |
| Microzooplankton mortality rate ($m_{Z_i}$; Daewel and Schrum (2013) ) (d$^{-1}$) | 0.4 | 0.16 |

et al. (2011):

$$\alpha_j(I) = \frac{(\frac{Chl_j}{P_j} * \alpha_{P_j} I(x,y,z,t))}{\sqrt{(\theta_{P_j}^{max})^2 + (\frac{Chl_j}{P_j})^2 * \alpha_{P_j}^2 * I(x,y,z,t)^2}} \tag{18}$$

Following these changes, chlorophyll *a* concentration has a direct influence on phytoplankton productivity, and our initial experiments suggested that the model was too productive compared to the observed values obtained from BGC-Argo profilers (results not shown) when using the REF parameterisation set. For this reason, the parameters relating to productivity, such as growth and grazing rates were modified in EXP simulations to keep the primary production level comparable with the observations. The assigned new values (Table 2) are similar to those given in Daewel and Schrum (2013). The changes can be summarized as decrease in phytoplankton growth rates, decrease in grazing rates to reduce pressure on phytoplankton with the new lower growth rates, and reduction in zooplankton mortality rates to balance the reduced grazing rates. For more details on the use of these parameters, see Section 3.3.2.

## 2.4 Along-track modelling setup

### 2.4.1 Preparation of forcing files

Along-track BGC-Argo modelling was conducted on 8 BGC-Argo trajectories. One experiment is performed separately for each trajectory, and the models are configured in separate folders. Several criteria were involved for the choice of trajectories suitable for the modelling experiments: (1) BGC-Argo chlorophyll *a* data should have enough resolution to represent temporal variations and high resolution changes at depth in order to validate model chlorophyll *a*. In the case of temporal resolution, Silva et al. (2021) gives a range of 28 - 58 days for the duration of the spring bloom for the Norwegian and Barents Seas. It is highly unlikely that conventional on-board in situ observations could provide the samples to cover the onset, peak and decay of the spring bloom within a large regional area, whereas with sampling frequency of 5 - 10 day, BGC-Argos can capture the changes

for the duration of the spring bloom in the Nordic Seas. BGC-Argo buoys with long-terms gaps in time were either avoided, or the years with missing data were not included in the experiment, (2) BGC-Argo buoys that were sampling a continuous and similar water mass in the same region during its course were selected to construct the suitable environmental conditions for the model (e.g. nutrient, temperature and salinity climatology), (3) BGC-Argo buoys with at least 1 year time-series data were chosen to represent a full-year cycle, and buoys with multiple years were prioritised, (4) although the Norwegian Sea is given a priority, buoys from other regions such as the south of Greenland or the North Atlantic Subpolar Gyre were selected for wider regional coverage. If a subsection of the whole BGC-Argo trajectory fit those criteria, only that time frame was included in the model

We set the model initial conditions using profiles representative of the BGC-Argo location data for both the physics (temperature and salinity) and nutrients (nitrate, silicate, and phosphate). WOA18 monthly nutrient, temperature and salinity were retrieved from the closest location to the buoy coordinate at the start of the simulation. A monthly time-series text file was prepared for each nutrient, temperature and salinity variable indicating year, month and the 15th day covering the years 2000 – 2020. The model automatically interpolates the data to the exact date of the model time. WOA18 data files were set to cover the whole water column. For each T and S profile, the upper 1000 m (due to most coverage and continuity) values were taken from BGC-Argo buoys, whereas depths below 1000 m were copied from WOA18 data interpolated to the exact coordinate of the buoy and date. This was to ensure realistic environmental conditions were present prior to the model spin-up. World Ocean Atlas 2018 (WOA18; Boyer et al., 2018; Locarnini et al., 2019; Zweng et al., 2019; Garcia et al., 2019a, b)) monthly climatology for temperature, salinity, nitrate, silicate, phosphate and oxygen were used as the initial conditions and monthly relaxation data.

High resolution (1-hour) ECMWF Reanalysis v5 (ERA5; Hersbach et al., 2020) was used to construct the atmospheric forcing along the buoy trajectory was applied to replicate the physical conditions at the ocean surface, and the surface shortwave radiation was included as an atmospheric forcing to drive the primary production by the phytoplankton. Latitude and longitude of the buoys were linearly interpolated to 1-hour intervals to precisely locate the closest point of atmospheric data for each forcing. The ERA5 products used to construct the atmospheric forcing were: (1) total cloud cover, (2) mean total precipitation rate, (3) mean surface net shortwave radiation flux, (4) mean sea level pressure, (5) 2-meter temperature, (6) 2-meter dew point temperature, and (7) 10-meter U,V wind component. These datasets were stored in a text file with a column for each, and an extra column for the time variable.

### 2.4.2 Model experiments

We experimented with two versions of ECOSMO II(CHL), the REF and the EXP respectively, with each having three generic set of simulations: (1) spin-up (spinup), (2) along-track relaxing to climatology (WOA) and (3) along-track relaxing to BGC-Argo (Argo). Each BGC-Argo track modelling experiment followed the common set of simulations depicted in Figure 3.

An along BGC-Argo track experiment starts with performing two-cycles of 8-year spin-up. The 8-year integration window is done to reduce the boundary condition size and preparation time of a longer-term atmospheric forcing. For the spin-up simulations, the model temperature, salinity and nutrients were weakly relaxed to monthly climatology (1-year relaxation time

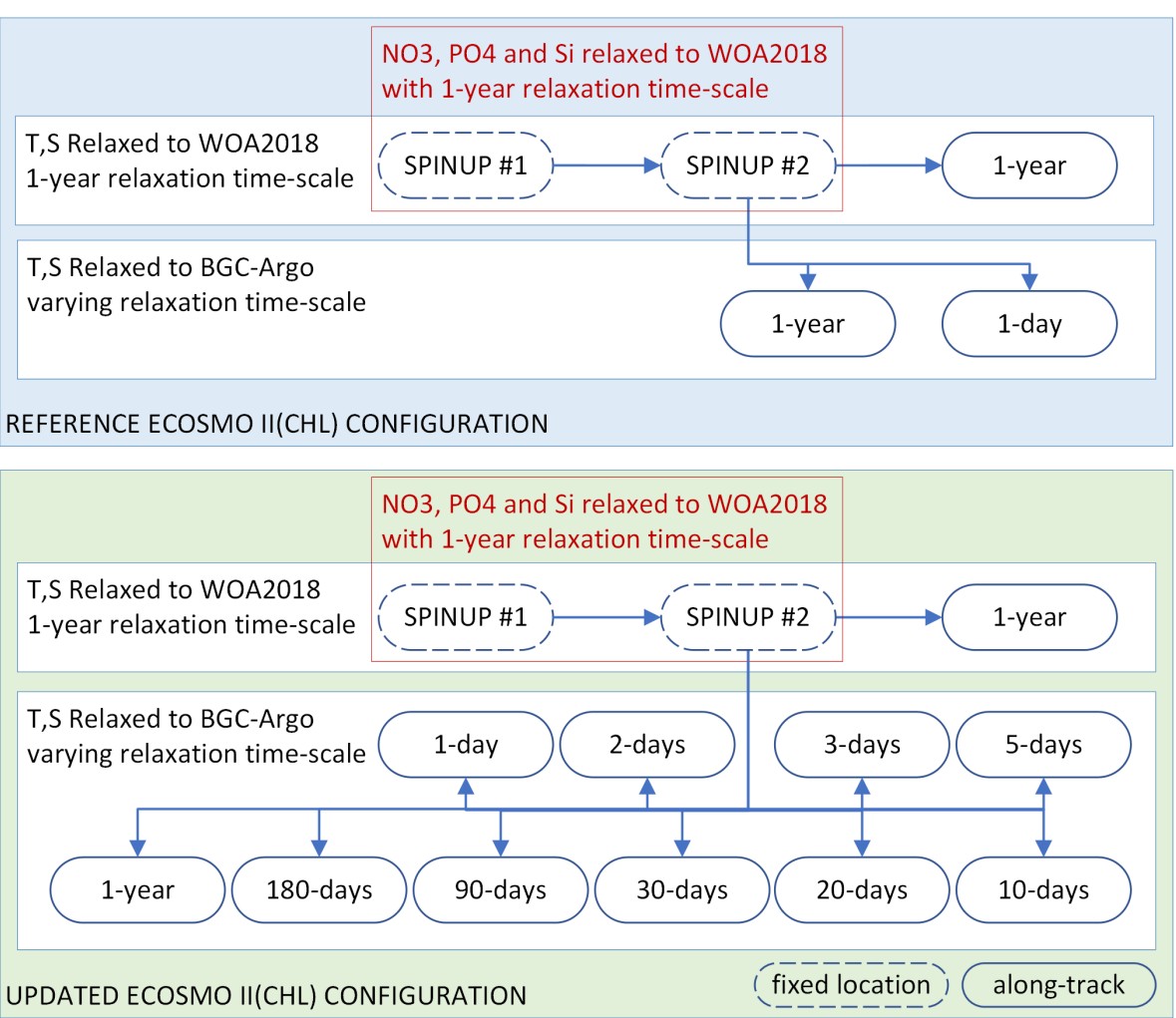

**Figure 3.** Summary of the two sets of simulations: (1) using the reference ECOSMO II(CHL) configuration (REF, blue box), (2) updates on the reference configuration (EXP, green box). The experiments rely on temperature (T) and salinity (S) relaxation to either WOA2018 or BGC-Argo data with varying relaxation time-scales where the spin-up simulations were relaxed to WOA2018 T and S at a fixed location which is the initial point of along-track simulations. Along-track simulations were conducted in parallel using the final day of the Spin-up #2 as the initial conditions. Spin-up simulations were also relaxed to WOA2018 NO3, PO4 and Si with a 1-year relaxation time-scale. No nutrient relaxation was applied to along-track simulations.

scale; $3 \times 10^7$ seconds). The spin-up simulations were performed at the coordinates of the first time-step of the along-track simulations, thus establishing a stable model initial condition. The nutrients were relaxed to climatology values to prevent drifts during a 16-years simulation. Since the spin-up model location is common to all experiments for a particular BGC-Argo, 205   it is simulated once and the restart file is stored for the along-track experiments.

Each along-track simulation starts with the initial conditions provided by the second spin-up. The along-track experiments differ from each other by the temperature and salinity datasets they were relaxed to (WOA18 or BGC-Argo) and the relaxation time-scale that was used in the simulation. To prevent artificial nutrient additions, relaxation to nutrients for both the 'WOA' and 'Argo' simulations were turned off. The experiments are identified by an abbreviation with prefixes indicating ECOSMO version, the BGC-Argo number (if necessary for the text), and the suffixes indicating the dataset they were relaxed to and the simulation relaxation length scale (e.g. REF-6902547-WOA-1year or in short: REF-WOA-1year when it is obvious from the text that the BGC-Argo number is 6902547).

The concept behind the experiment setup depicted in Figure 3 is as follows:

1. The experiments are divided into two major groups: (1) REF, the reference ECOSMO formulation and parameterisation of Yumruktepe et al. (2022b) and (2) EXP; the final formulation and parameterisation set after a series of experiments conducted during this study. The sensitivity analyses conducted to achieve the EXP parameter set is not presented here. In the following sections, the REF and EXP experiments are compared and the improvements and shortcomings of the EXP experiments are discussed.

2. Relaxing a 1D model to a climatology dataset is a very common practice. Thus, for each REF and EXP category, the 'WOA' simulations are used as the reference simulations to evaluate the added value of relaxing the model T and S to the BGC-Argo T and S with the assumption of reducing errors in modelled physics.

3. The 'Argo' simulations, each of which having a unique relaxation timescale showcasing the effect of the strength of re-laxation towards BGC-Argo T and S on the biogeochemistry. Comparing each 'Argo' simulation to the respective 'WOA' simulation, and comparing them all, including the 'WOA', to the BGC-Argo chlorophyll *a* would yield the performance of our modelling approach along the BGC-Argo track. Performing an analysis on 'WOA' and 'Argo' simulations' physics would determine the optimal relaxation time-scale as a reference for future studies of the similar approach.

4. The best performing 'EXP' simulation's formulation and parameterisation would be the final outcome of our approach, and subject to further testing in a 3D modelling framework, for a potential upgrade to the ECOSMO II(CHL) model formulation.

## 2.5 Model statistical analysis

When constructing the statistical evaluation of the model along the BGC-Argo track, the BGC-Argo sample points were linearly interpolated to model depth. The model and BGC-Argo data are separated into monthly and 10-meter depth interval clusters. A statistical analyses is performed for each cluster and the bias and rmse was calculated following Eqs. 1 and 2 respectively (See Section 3.3.3). The statistics are calculated for chlorophyll *a* in $[\log_{10}(\text{mgChl m}^{-3})]$ units.

## 3   Results and discussions

### 3.1   Biogeochemical Argo data evaluation in the Nordic Seas

BGC-Argo chlorophyll *a* data has been receiving quality checks and adjustments (e.g. Xing et al. (2012); Roesler et al. (2017)) and has improved significantly over the recent years. A central adjustment is the division by 2 suggested by Roesler et al. (2017). They suggested that this division improves the overestimation of the factory calibrated chlorophyll *a* estimates for WET Labs Environmental Characterization Optics (ECO) series chlorophyll fluorometer sensors. However, this adjustment is a global average correction where regional value may differ and before progressing further with the model experiments, it is therefore important to evaluate the BGC-Argo chlorophyll *a* data for the Atlantic north of $50^o$N, and evaluate whether the division by 2 is also valid for that region.

Evaluating the BGC-Argo chlorophyll *a* against in situ data would have been the preferred choice as these samples cover deeper layers in the water column. However, too few co-locations between BGC-Argo and in situ samples were present (Fig. 1) leading to an unreliable statistical analysis. Therefore, satellite chlorophyll *a* data was used as an independent cross-validation dataset, and BGC-Argo and in situ sample chlorophyll *a* were separately analysed statistically against satellite data. We note that van Oostende et al. (2022) shows inconsistencies within the continuity of OC CCI v5.0 chlorophyll *a* product appearing as sudden steps in the time-series. These steps appear when a satellite is launched or removed. For this reason, we limited our statistical analysis from May 2012 to May 2016 where only MODIS and VIIRS are continuosly active, as this time frame fits our study period. Figure 1 in van Oostende et al. (2022) depicts no sudden steps in the OC CCI V5.0 data for this period.

Both visually and statistically (Fig. 4), the BGC-Argo and in situ sample chlorophyll *a* is generally similar to the satellite chlorophyll *a*. This analysis indicates that the default quality corrections applied for BGC-Argo chlorophyll *a* ensures a good representation of the in situ sample chlorophyll *a*, while noting that satellite data is limited to the optical depth at the surface. Unfortunately, we do not have the required data at depth to compare the BGC-Argo data to and we rely on the quality information document for the reprocessed in situ observations (Jaccard et al., 2018). Following this evaluation, we conclude that BGC-Argo chlorophyll *a* dataset is of sufficient quality to be used as a validation tool for biogeochemical models for the Nordic Seas without any further post-processing. This evaluation ensures that we can proceed with the modelling experiments.

### 3.2   Along-track model physics evaluation

The model physics and its variations stemming from different relaxation scales play a crucial role for the biogeochemistry and we therefore evaluated the effectiveness of the different relaxation scales on the temperature and salinity profiles along the tracks. By simulating temperature and salinity as similar as possible to the observed values (within a margin allowing the model dynamics the freedom to perform properly), we establish the foundation for the biogeochemical model experiments that minimize the impact of errors stemming from model physics. As a result, we can target the biogeochemistry for improvements.

In our approach, we considered the simulation where model T and S were relaxed to WOA18 with 1-year relaxation time scale (e.g. REF-WOA-1year) as a typical setup for a 1D simulation, and can thus be considered a reference experiment. This

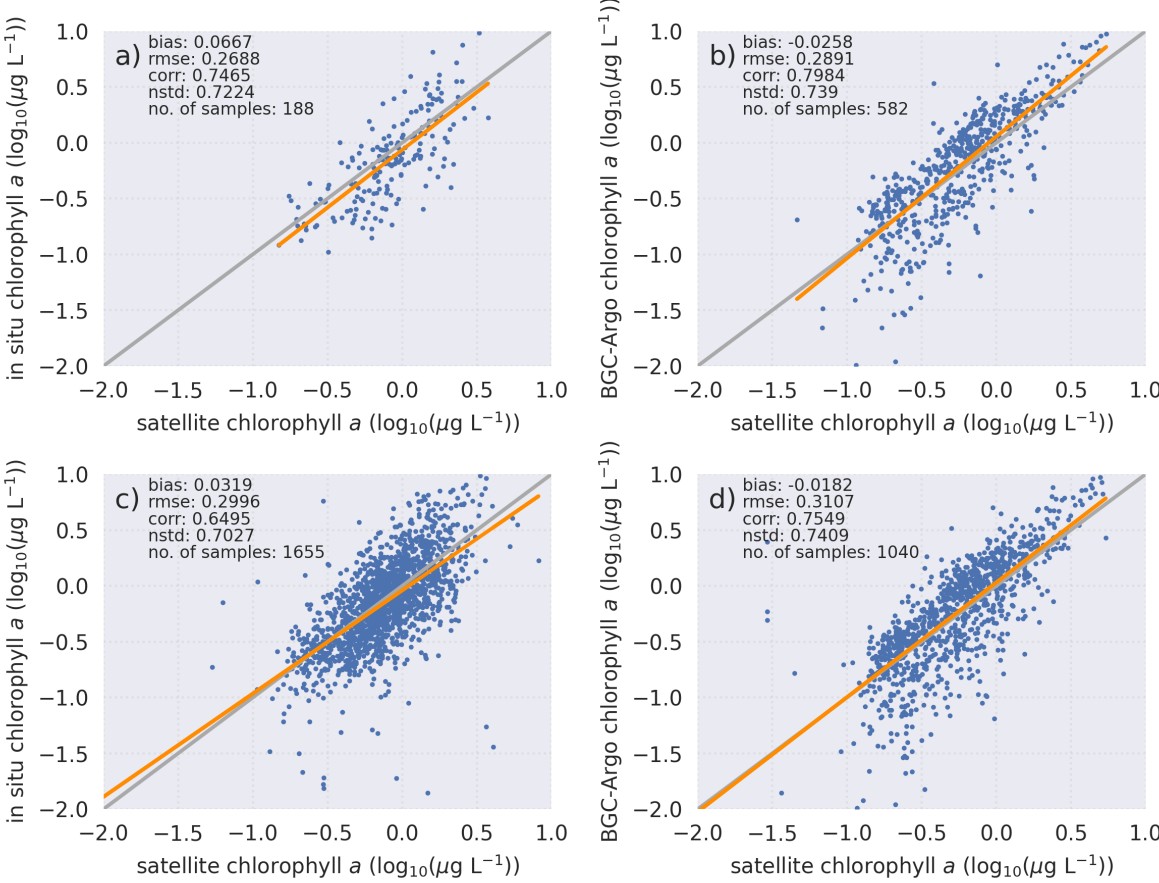

**Figure 4.** Chlorophyll *a* statistical analyses of in situ bottle samples with a search radius of (a) 2 km, (c) 10 km and BGC-Argo with a search radius of (b) 2 km, (d) 10 km reveal that BGC-Argo statistics against satellite chlorophyll *a* show the same pattern as in situ bottle statistics against satellite chlorophyll *a*. The computed statistics and number of sample points for each sample set is depicted in the figures. Equations for the computed statistics are described in Sec. 2.2. Data from all sources are $\log_{10}$ transformed.

is the case for both the model physics and the biology. The remaining experiments (i.e. REF-Argo-'relaxation_scale') are the iterations that allows us to evaluate the optimal relaxation time scale for progressing with the biogeochemical experiments.

The simulations REF and EXP can be used interchangeably as they both use the same physics.

We use BGC-Argo 6902547 (Fig. 5) to showcase the capabilities of the framework we designed, but the remaining experiments' figures are included in the supplementary material. We use BGC-Argo and satellite sea surface temperature to evaluate the simulated temperature, salinity and mixed layer depth (MLD) from the experiments with different relaxation scales (Figs. 6, 7 and 8). For a visual clarity, Fig. 8 focuses on a portion of the BGC-Argo track with a limited number of simulations. The

275 full time-period and set of simulations are depicted in Figure A1.

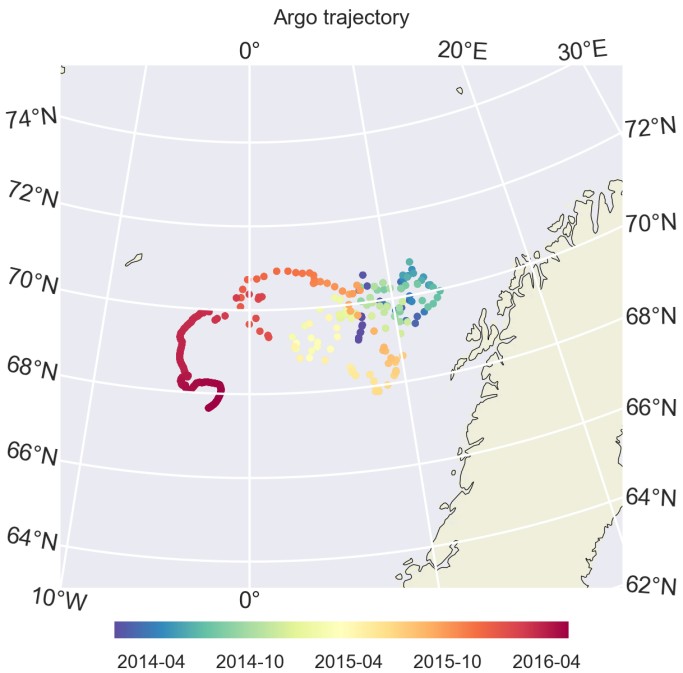

**Figure 5.** During its drift, BGC-Argo 6902547 was located in the Norwegian Sea confined and was mostly trapped in the Lofoten Basin. Later in 2016, it drifted westwards. We limit our model evaluation prior to this period and mainly focus on 2014 and 2015.

Both the simulated temperature and salinity (Fig. 6) were progressively more similar to the observed values with shorter relaxation time scales (i.e. stronger influence of BGC-Argo T and S profiles), where the simulation with WOA18 relaxation has the least short-term variability. Even in the case of 30-days relaxation (REF-Argo-30days), temporal variability is less pronounced compared to that of the 1-day relaxation (REF-Argo-1day). An argument can be made where evaluating the model
results with the dataset that it was relaxed to may raise concern, but when the simulations were compared to an independent dataset (i.e., the satellite sea surface temperature (SST; Fig. 7)), the differences among the experiments are evident, more for the months between October - May where vertical mixing is high. REF-WOA-1year has the lowest performance for these months, and it requires 30 days or less relaxation time scales to perform a better fit with the observed SST, while the short-term relaxations (1 - 5 days) performs better.

We have a particular focus on MLD as it is important for controlling phytoplankton phenology (e.g. timing, depth and duration of bloom events). All the simulations use the same atmospheric forcing, therefore the choice of relaxation dataset and the time-scales were the dominant drivers of differences in MLD. Notably, MLD (Figs. 8 and A1) for both cases of 1-year relaxation scales, REF-WOA-1year and REF-Argo-1year were similar and were consistently calculating MLDs deeper than the observed, especially for 2014 and early-spring of 2016. This is also evident in Figure 7 where SSTs for these two
simulations for the mixing period are cooler compared to the other experiments and the observed values. Due to a better fit with the observed MLD estimate, the short-term relaxation scale experiments (30 days or less) achieve a more pronounced

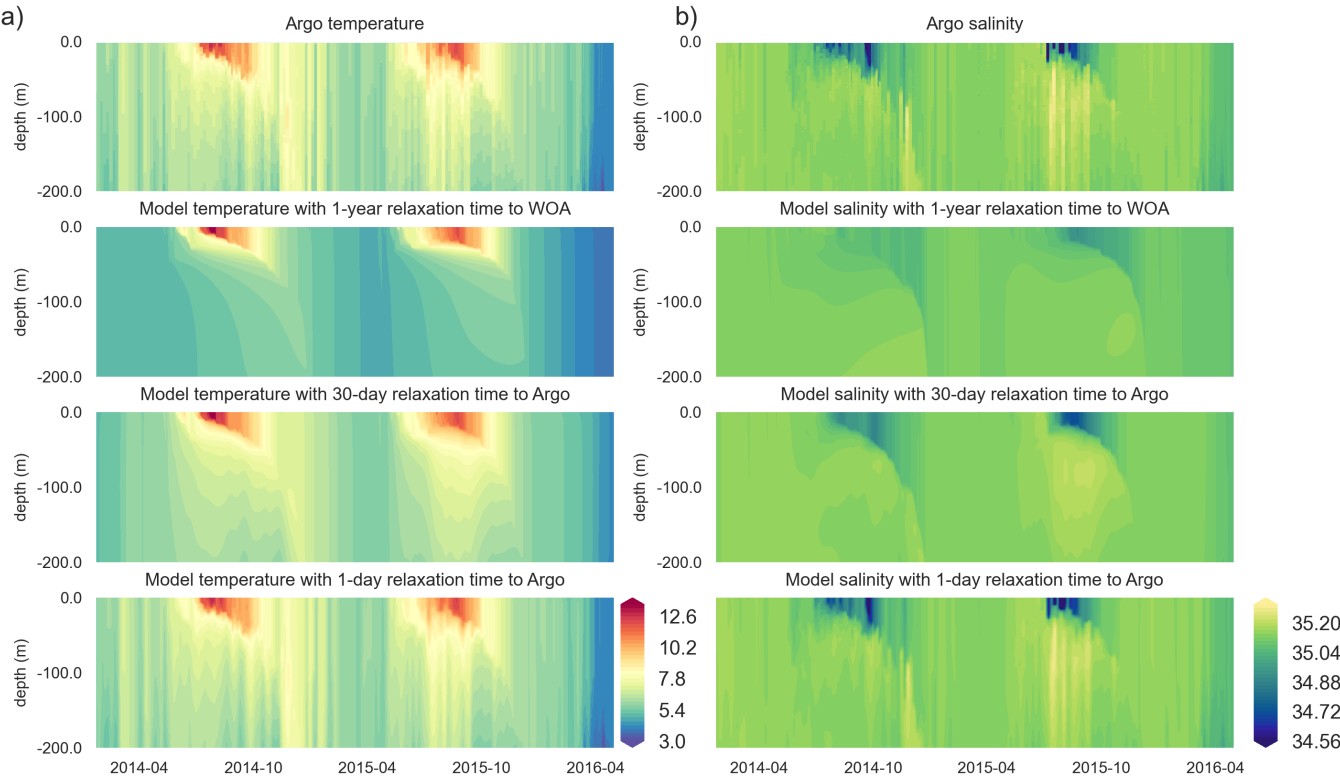

**Figure 6.** Along BGC-Argo 6902547 track vertical temperature (a) and salinity (b) Hovmöller plots depict increasing similarities between the BGC-Argo and modelled temperature and salinity as the relaxation time scale parameter decreases, to practically identical values when 1-day relaxation scale was used.

inter-annual variability in MLDs (e.g. shallower winter MLDs for 2014 and deeper for 2016). These results demonstrate that the use of Argo-driven model physics produce a more optimal physics for the biogeochemical simulations. For this purpose, as we progress with the model results in the following sections, we will be focusing on the model results of the 1-day relaxation scales experiments when showcasing the relaxing to the BGC-Argo experiments.

### 3.3 Modelled chlorophyll *a* evaluation

#### 3.3.1 Evaluation of the reference ECOSMO II(CHL) formulation

We first evaluate the reference ECOSMO II(CHL) simulations in order to detect shortcomings, and later focus on these to improve the model results by objectively analysing them against the BGC-Argo chlorophyll *a*. This approach and its outcome is the primary objective of our study.

We present the observed (Fig. 9a) and simulated (REF-WOA-1year and REF-Argo-1day; Figs. 9b and c respectively) chlorophyll *a* along the same BGC-Argo trajectory described in Section 3.2. We used the REF-Argo-1-day for this comparison since

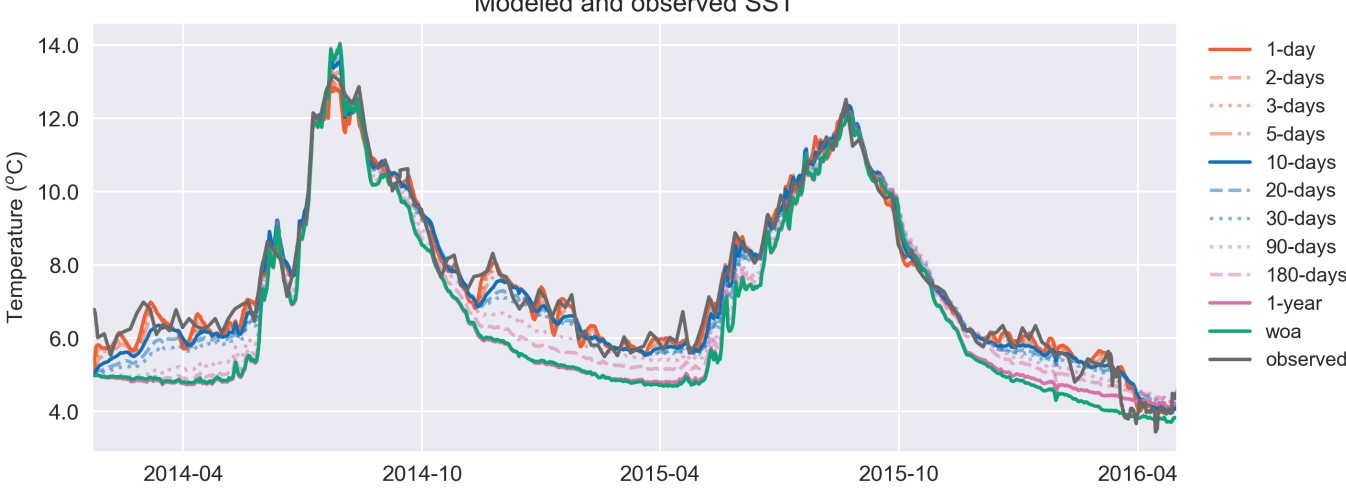

**Figure 7.** Similar to T in Figure 6a, the model surface temperature similarly performs better against satellite sea surface temperature as the relaxation scale decreases. The simulation identified as 'woa' corresponds to REF-WOA-1year and the remaining simulations correspond to REF-Argo- with different time-scales of relaxation.

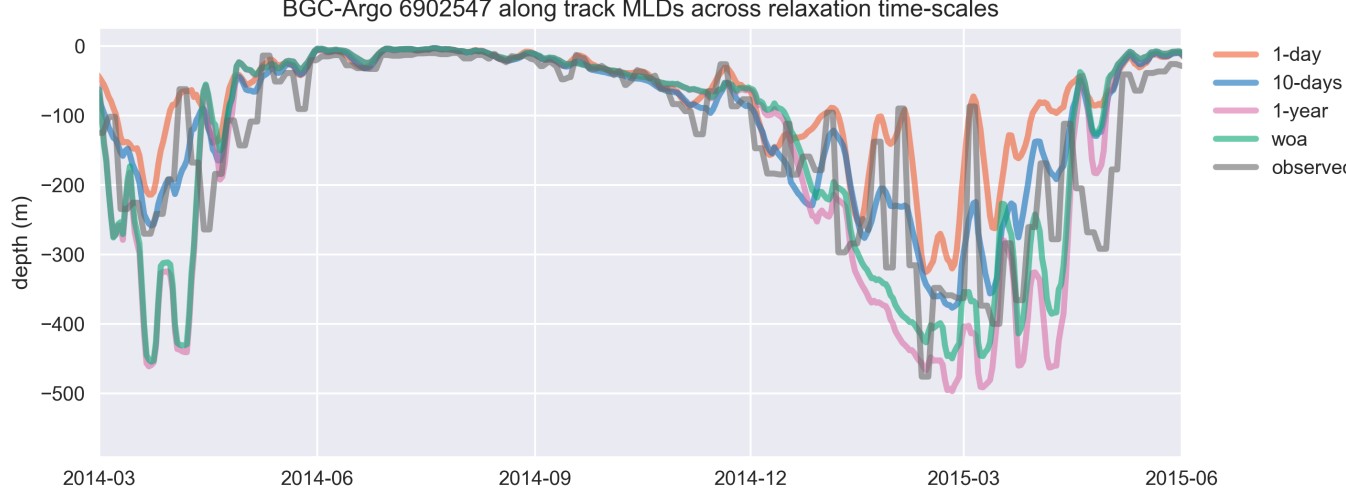

**Figure 8.** Mixed layer depth (MLD), estimated by using 0.03 kg m$^{-3}$ density change criteria from 10 meters, similar to the cases of temperature and salinity, is represented better in experiments that use shorter relaxation time scales. The model MLDs depicted in this figure are GOTM model outputs 'MLD_surf' calculated from turbulence. For visual clarity, the time period of the BGC-Argo track is shortened, and model experiments are limited to a few representative simulations. The full time-period including every simulation is provided in Figure A1. The simulation identified as 'woa' corresponds to REF-WOA-1year and the remaining simulations correspond to REF-Argo- with different time-scales of relaxation.

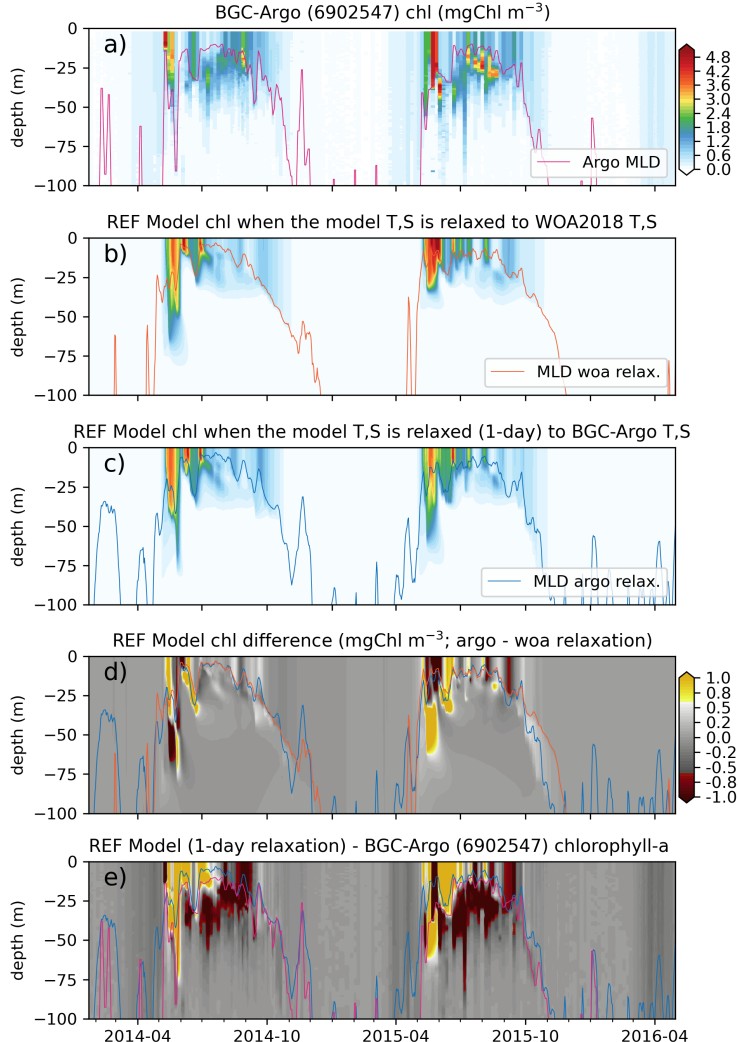

**Figure 9.** Comparison between (a) observed (BGC-Argo 6902547) and (b) REF-WOA-1year and (c) REF-Argo-1day modelled chlorophyll *a* reveal notable differences between the two models (d; REF-Argo-1day - REF-WOA-1year) and between the model and observed data (e; REF-Argo-1day - BGC-Argo). We note that for panel e, the BGC-Argo chlorophyll *a* was linearly interpolated to the model depth points.

the MLD, T and S were better represented with a short-term relaxation time-scale (Figs. 6, 7 and 8). Investigating the optimal time-scale for relaxation (see Section 3.4) reveals that the 1-day relaxation provide the statistically best results, although

differences between the short-term (1 - 10 days) relaxation time-scales are minor and are subject to the requirements of the modelling study.

The observed chlorophyll *a* shows relatively minor increases in concentration (<0.5 mgChl m$^{-3}$) in early March for the three years with the first peak-bloom of the year during early-May. The first peak in 2014 was followed by a decrease in concentration and forming a second peak (though weaker) in late-June 2014. Following the second peak, the data shows the

formation of a deep chlorophyll *a* maximum (DCM) around 20 – 50 meters depth range. A comparable second peak in June 2015 was not observed, but like in 2014, a DCM layer was formed at similar depths. The formation of the DCM started earlier in 2015. For both 2014 and 2015, a late-summer (September - October) increase in chlorophyll *a* concentration from surface to deeper than 75 meters were observed. During the spring bloom events, the observed chlorophyll *a* concentrations exceeded 5 mgChl m$^{-3}$ and in the case of 2015, DCM concentrations exceeded 4 mgChl m$^{-3}$. Such continuous and high resolution vertical, seasonal and inter-annual variability in the observed data showcases the unique value of BGC-Argo observations for model evaluation. Especially for the vertical case, BGC-Argo buoys are often the only source of available observation in the open ocean.

The two simulated chlorophyll *a*, REF-WOA-1year (Fig. 9b) and REF-Argo-1day (Fig. 9c), have in general similar vertical and temporal patterns and concentrations. Their first peaks occur in May - June for 2014 and 2015 followed by lower concentrations in summer. However, there are notable differences (REF-WOA-1year is subtracted from REF-Argo-1day; Fig. 9d) between the two experiments. In addition, the differences are not consistent between the simulated years. The timing and depth of the differences vary between 2014 and 2015. For 2014, the REF-Argo-1day chlorophyll *a* concentration is higher at the surface during spring bloom (May) while REF-WOA-1year chlorophyll *a* in response is higher below 40 meters. This pattern for May - June is reversed for 2015. REF-WOA-1year chlorophyll *a* is higher at the 0 - 25 m range and REF-Argo-1day is higher below. During July of both years, REF-Argo-1day chlorophyll *a* is higher for the surface to 25 m depth range. For this depth interval after July, REF-WOA-1year simulates higher peaks in chlorophyll *a* concentration during August - September 2015. Although minor ($\sim$ 0.5 mgChl m$^{-3}$), REF-Argo-1day simulates higher chlorophyll *a* concentrations below MLD. The difference is prominent during October and is located as low as $\sim$ 100 m depth. Although not as prominent as October, during early May the REF-Argo-1day chlorophyll *a* is slightly higher reaching $\sim$ 100 m depth.

These earlier increases in chlorophyll *a* concentrations in May due to modified model T and S may be attributed to the earlier shoaling of the MLD in REF-Argo-1day case (Fig. 8) which is in better agreement with the estimated BGC-Argo MLD. Shallower MLDs may decrease the light-limitation on phytoplankton growth and thus allows the conditions suitable for growth earlier compared to the REF-WOA-1year case. At the end of the growth season, during October (especially in 2015), a deeper MLD in the REF-Argo-1day case (in better agreement with the estimated BGC-Argo MLD) allows for a larger intrusion of nutrients towards the nutrient-limited surface layers. This allows more productivity for these late summer periods which is prominent in $\sim$ 25 - 100 m depth range. These noted differences correspond to the changes in biology when the model T and S is altered by strongly relaxing them to the BGC-Argo T and S, and showcase the changes in biology with respect changes in model physics only.

After minimizing the physics model errors, and the resulting errors in biology, we can identify and target the differences in simulated and observed chlorophyll *a*. To identify the differences, we compare the estimated chlorophyll *a* of REF-Argo-1day simulation to the observed chlorophyll *a* from the BGC-Argo. In a similar fashion to Figure 9d, observed chlorophyll *a* is subtracted from the REF-Argo-1day chlorophyll *a* (Fig. 9e). With this comparison, we detect important patterns of differences: (1) The model fails to reproduce a distinct deep chlorophyll maxima (DCM), as the difference is always on the high-negative scale throughout June - September in the 20 - 50 m depth range (sometimes as deep as 75 m), and is on the high-positive scale

near the surface, (2) the timing of spring bloom initiation is late as the difference is on the low-negative scale during April -
May, which is consistent with the simulated shallower MLD during this period.

### 3.3.2   Phytoplankton growth formulation and parameterisation

Prior to discussing the changes to the model, it is important to elaborate on the effect of uncertainty of the BGC-Argo data, for
we rely on this dataset to exert changes to the model code and parameterization. As is the nature of observations, they all are
different than the true value, and there will be mismatches (Skogen et al., 2021), even in the case of in situ chlorophyll *a* bottle
samples. Nevertheless, while we acknowledge that there are mismatches among different datasets (Fig. 4), we can still retrieve
enough information from the BGC-Argo dataset to detect model shortcomings and propose improvements. For example, in
every case where the model was nudged towards the BGC-Argo temperature, stronger relaxations result in a better match
between the model T and SST which is an independent dataset to BGC-Argo (Fig. 7). Similarly, in the case of BGC-Argo
chlorophyll *a* uncertainty, we are not pursuing a precise 1-to-1 match between the model and BGC-Argo, but exploring notable
differences that should be improved regardless of the concentration differences. As such, there are fundamental errors in the
model that need to be addressed, i.e. the late-bloom which disrupts the timing of energy transfer to the upper trophic levels and
the absence of DCM which is the production that is not accounted for in the model. These fundamental dynamics are observed
in the BGC-Argo data even if they may not be represented by precise accuracy. Therefore, in the experimental phase, we focus
on these two issues and investigate ways to improve the mechanics of the model in general. Noting these, fine-tuning model
parameters in a follow-up study would require a more research on the effect of BGC-Argo data uncertainty.

There are various hypotheses on the effect of mixing or stratification on the initiation of the spring bloom in the north
North Atlantic from the "critical stratification threshold" (Sverdrup, 1953) where sufficiently abundant light due to shallower
convective mixing allows the growth to exceed losses, to the 'dilution–recoupling hypothesis' where deep winter mixing dilutes
prey and predators, thus decoupling phytoplankton growth and grazing loss rates by reducing encounter rates (Behrenfeld,
2010). Later during the spring stratification, phytoplankton and zooplankton recouple with enhanced growth rates due to light
abundance and grazing rates due to increased encountering. For the Nordic Seas, Mignot et al. (2016) suggests the photo-
period (the number of daily light hours experienced by the phytoplankton) exceeds a critical value. Common to all of these
hypotheses is that stratification plays an important role on pytoplankton productivity, and errors in model physics can thus be
an important source of errors in the modelled biogeochemistry. However, we have shown the reduction in the model physics
errors by relaxing the model T and S to the observed values in Section 3.2 and documented the changes in biogeochemistry in
Section 3.3.1. This suggests that the major source of error in phytoplankton growth is related to the biogeochemical model.

Also common between these hypotheses is the critical importance of light abundance. Even in the case of Behrenfeld (2010),
though focusing more on prey/predator interactions, light has a central role. During periods where phytoplanton are decoupled
from zooplankton, the phytoplankton are still dependent on light to sustain growth. However, neither REF-WOA-1day nor
REF-Argo-1day simulations reproduce the winter biomass detected by the BGC-Argo (Figs. 9a and e). Both the late bloom
and the absence of a DCM suggests that the modelled phytoplankton growth is too low under low-light conditions. This

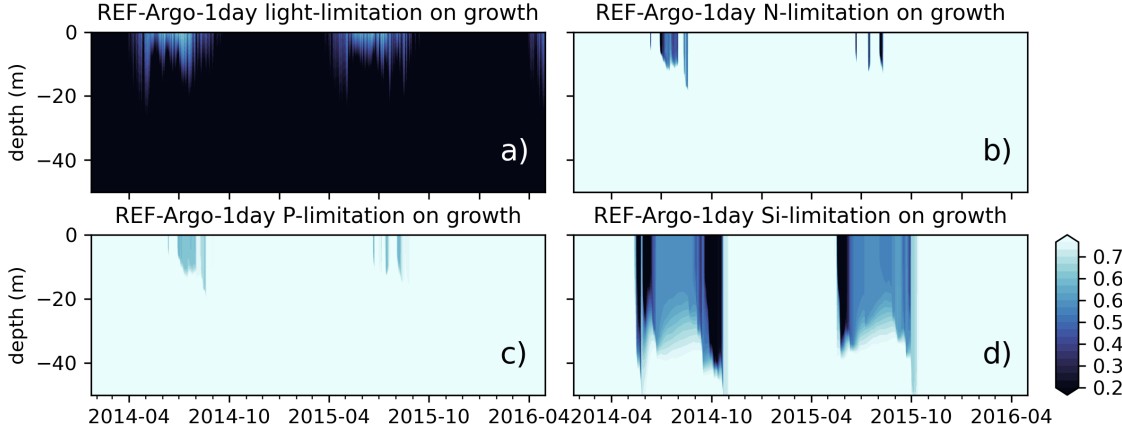

**Figure 10.** Simulated (a) light-, (b) nitrogen-, (c) phosphate-, and (d) silicate-limitation on growth shows that the model is mostly light-limited which can partly explain the low chlorophyll *a* during early spring or summer below the surface. The BGC-ARGO chlorophyll *a* (Fig. 9a) forms a DCM between 15 - 50 m depth range during the summer months, whereas the model is light-limited for this depth interval. The model effectively simulates silicate limitation during the spring and late summer blooms allowing a switch in dominance from diatoms to flagellates (results not shown). During summers, surface layers are nutrient limited, mainly nitrogen. Note that the limitation on growth is a value between 0 and 1, where for visual purposes, we have limited the color range to 0.25 - 0.75. Darker colors suggest limitation.

evidence is supported by the model growth limitation results (Fig. 10) where during low-light conditions, the growth is light limited.

A prime candidate for the growth limitation due to low-light could be the high light attenuation, but is unlikely in this case as the low-growth occurs throughout the year, hence in the case of late-bloom, there is not enough phytoplankton to cause excess self-shading. This suggests that the model phytoplankton are not optimally utilising the available light (PAR). The default (REF experiments) ECOSMO II(CHL) formulation on light-limitation (Eq.9) defines the light-limitation as a hyperbolic tangent curve with a photosynthesis efficiency multiplier. This function is later multiplied by the maximum growth

rate, but it does not introduce variability to various conditions (e.g. light intensity, internal cellular structure). The strength of growth is moderated by the efficiency constant which can increase/decrease the productivity as a whole rather than introducing seasonality or variations at different depths.

To achieve a certain level of variability in the EXP simulations, we have introduced a more dynamic light-limitation on growth (Eq.18) that takes into account the C:Chl ratios which is defined as a function of light intensity. This formulation

introduces enhanced productivity in cases of high intra-cellular chlorophyll *a* content, which the model reproduces for low-light conditions. By introducing this functionality, we have disrupted the fine-balance of the model parameterisation, and we therefore needed to modify some of the parameters related to phytoplankton growth, and in connection, the grazing rates. The aim with this study is not to fine-tune the model parameters, but to showcase the capability of using the BGC-Argo buoys as tools for model improvements. Fine-tuning the model parameters requires a cluster of model experiments and multi-regional

representations where here we focus on a limited number of BGC-Argo tracks. We have performed a series of experiments on

model parameters to present the added value to the model, and show how the model can be objectively validated using the BGC-Argo data. Table 2 summarises the changes to the parameters, which lead to a decrease in the the maximum phytoplankton growth rates. We decreased the grazing pressure of the zooplankton on the phytoplankton to balance this change. Due to the lower zooplankton food intake, we also decreased their mortality rate to sustain zooplankton biomass.

### 3.3.3 Evaluation of the updated ECOSMO II(CHL) formulation

The results for the experiments described in Section 3.3.2 are presented in Fig. 11. The simulated chlorophyll *a* (Fig. 11a) shows bloom initiation in April for both 2014 and 2015 reaching the peak concentrations in May, followed by consecutive decreases and increases in concentration at the surface throughout the summer with prominent DCMs within 20 - 50 meters depth interval during July - September. Similar to the previous simulations, after the DCM period, with the increase in MLDs, late summer peaks occur as deep as 75 meters in October. At its highest, the chlorophyll *a* concentration is $\sim$5 - 6 mgChl m$^{-3}$.

While in general the simulated chlorophyll *a* is similar to those described in Section 3.3.1, there are notable structural differences between EXP-Argo-1day and REF-Argo-1day (REF-Argo-1day is subtracted from EXP-Argo-1day; Fig. 11b), (1) the initiation and the peak of the spring bloom is earlier and (2) the summer subsurface chlorophyll *a* is more prominent in the EXP-Argo-1day simulation. As the spring bloom occurs earlier, the grazing pressure and the nutrient limitation (not shown) also initiates earlier. Both increasing the grazing pressure and increasing nutrient limitation results in decreased chlorophyll *a* concentration. During the chlorophyll *a* concentration decrease in EXP-Argo-1day, REF-Argo-1day is experiencing its peak bloom. These misalignment in chlorophyll *a* concentration peaks depicts a consecutive high positive/negative differences (Fig. 11b). Similar positive/negative differences appear throughout the summer at the surface suggesting that phytoplankton growth and loss imbalance have shifted earlier resulting in mismatches in local peak concentration timings. Below the surface (20 - 50 meters), during July - September, EXP-Argo-1day simulates higher chlorophyll *a* with lower values towards the surface, suggesting the presence of DCM. The late summer surface chlorophyll *a* concentrations are also higher in EXP-Argo-1day.

The EXP-Argo-1day and BGC-Argo chlorophyll *a* differences (BGC-Argo is subtracted from EXP-Argo-1day; Fig. 11c) suggest that the applied changes did not exert a perfect fix to the model. The results still depict differences on the high positive/negative ends. However, when comparing these to the REF-Argo-1day and BGC-Argo differences (Fig. 9e), we can see that the high-negative bias throughout June - September for 20 - 50 m depth range (See Section 3.3.1) pattern been reduced, as Fig. 11c depicts highs and lows for the 20 - 50 meters depth interval indicating increased phytoplankton growth at lower light conditions. For better accuracy, the timing of peak concentrations in the DCM layer should be improved in future tuning of model parameters.

In addition to the visual comparison, the statistical analysis provides an objective evaluation of the model results (Fig. 12). The high model bias and rmse during March - April have significantly decreased in the EXP-Argo-1day simulation attributed to an earlier (in better agreement with the observed values) bloom. Similarly, subsurface bias and rmse are improved during July - October, especially below 50 meters. These improvements suggest that the model is now able to represent effective growth during low-light conditions because the bias improvements to the high negative ends shows that the previously very low chlorophyll *a* concentrations have increased for the low-light conditions, which was our primary target for model improvement.

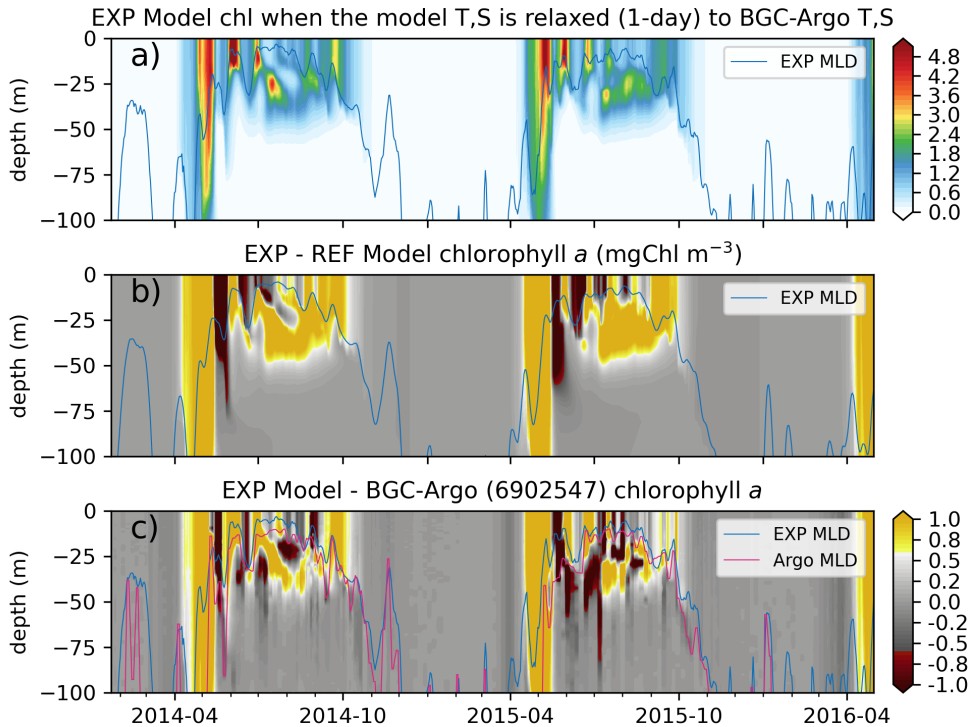

**Figure 11.** Updated ECOSMO II(CHL) formulation simulates (a, EXP-Argo-1day) chlorophyll *a* notably different to (b) REF-Argo-1day. Updated formulation improves the timing of the spring bloom and the summer DCM formation as the simulated chlorophyll *a* is higher during these events. The opposite patterns in Fig. 9e (i.e. model chlorophyll *a* is higher at the surface and lower below suggesting the absence of DCM) is weakened with the updated formulation, as such the simulated and observed differences in chlorophyll *a* depict (c) local highs and lows at the observed DCM depth. The differences are also reduced for the spring bloom period suggesting an earlier simulated bloom which is an improvement for the model. We note that for panel c, BGC-Argo chlorophyll *a* was linearly interpolated to the model depth points, thus its vertical resolution was decreased.

The simulated chlorophyll *a* still has flaws including occasionally being too high. For example, although the timing of the spring bloom has improved, the peak concentrations are high for April - May (Fig. 11c). The late summer concentrations during September - October are also high which is more pronounced in 2014. Although the model occasionally produces higher DCM concentrations, the differences there can partially be attributed to the dislocated depth positioning of the observed DCM. For example, during July - August 2014 (Figs. 9a and 11a), the observed DCM location ascends from below of 25 m to above

whereas the model DCM location descends from above of 25 m to below. The model does not produce a DCM during mid-June - early July 2015, thus having a negative difference. However, similar to the case in 2014, from mid-July - September there is a mismatch in the depth location of the DCM. Apart from these major differences, the model has shifted from a negative bias at the surface to a positive bias (more prominent within 0 - 40 meters).

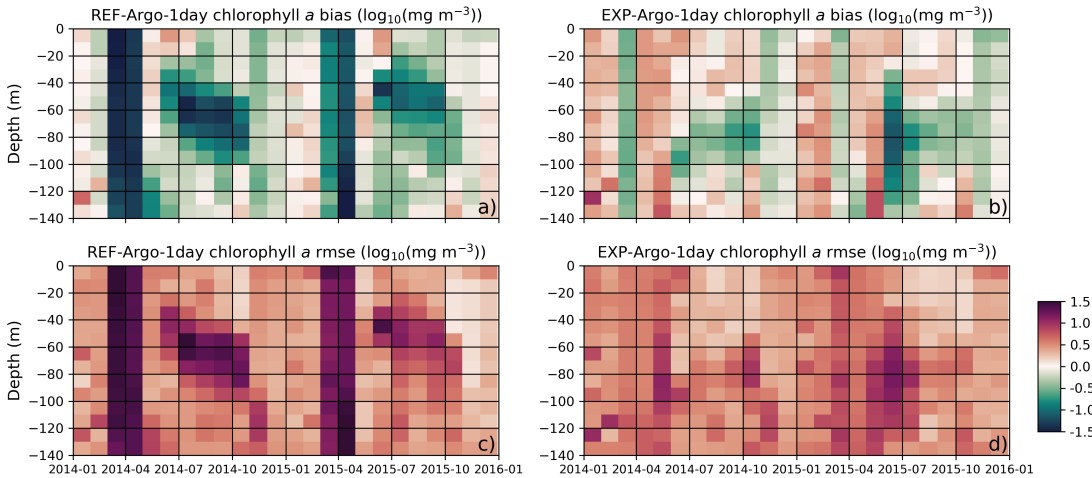

**Figure 12.** The reference model bias (a) has improved by including chlorophyll *a* in the light-limitation function, as such in a light-limited environment where the model produces a lower C:Chl ratio effectively decreasing the light-limitation on phytoplankton growth rates. In conditions where the light is limited, such as the winter/early-spring or the summer period subsurface, the model bias (b) received the highest improvements with the modified model. Similar to bias, rmse for the reference model (c) has improved the highest in the aforementioned conditions with the modified formulation (d). To construct these statistics, the BGC-Argo sample points were linearly interpolated to model depth and monthly averages were taken for 10-meter interval along the water column. The statistics are calculated for chlorophyll *a* in [$\log_{10}(\mathrm{mgChl\ m^{-3}})$] units.

To improve the clarity of our experimental study and the reasoning behind our thought process, we only focused on the BGC-Argo 6902547 in the main text of this paper. This way, we presents how one can approach our framework for model evaluation and improvements. However, an argument can be raised that our changes to the model formulation and parameterisation is case specific, i.e., the BGC-Argo 6902547 which is located in the Norwegian Sea between 2014 - 2016. In response to this argument, we point out that we targeted the phytoplankton growth under low-light conditions as we concluded was necessary for the model as the experiments revealed (See Section 3.3.2). The parameter changes given Table 2 represent parameter tuning to adapt the model to its new formulation on growth. Thus, although the prescribed parameters target the specific case of the BGC-Argo 6902547, the formulation change for low-light conditions should benefit the model in general, which is the case for other experiments we conducted in this study. We did not include the results of these experiments here, but the alternatives to Figures 6, 7, 8, 9, 11 and 12 for each specific experiment are provided in the Supplementary Materials. All reference experiments (REF simulations) are mostly light-limited, and every modified model experiment (EXP simulations) has objectively improved for the light-limited conditions according to the assessment statistics (see Supplementary materials).

These experiments are not only located in the Norwegian Sea, but cover various regions in the northern North Atlantic (Fig. 1), showing that the changes can improve model results throughout the northern North Atlantic which is the main region that ECOSMO II(CHL) is actively developed for (Yumruktepe et al., 2022b). As discussed above, fine-tuning is necessary as the model is too productive which is also the case for other experiments presented in this study. However, fine-tuning of parameters

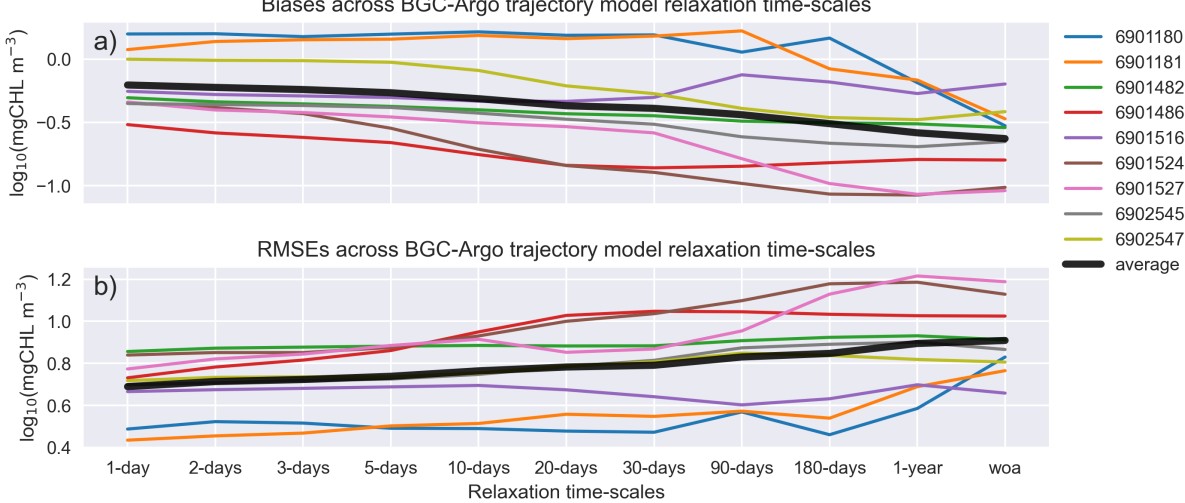

**Figure 13.** EXP-Argo experiments mean estimated March - May (a) bias and (b) rmse for different T and S relaxation time-scale to BGC-Argo. With varying degrees, the improvements are common to all the along-track experiments.

is beyond the focus of this study as it requires detailed sensitivity analyses. Here we have focused on building a framework where a BGC-Argo can be easily used for driving model physics and at the same time provide a high-resolution dataset for its evaluation. This paper is an example of how one can approach this framework. A follow-up study is needed to fine-tuning the model parameters using multiple BGC-Argo tracks (also updating the number of tracks with the most recent BGC-Argo deployments) in order to support model improvements on a 3D-model domain.

### 3.4 Discussion on the relaxation time-scales

An important issue which we have briefly discussed (See Section 3.2) is the choice of the length of the relaxation time-scale for T and S. Referencing the T, S and MLD comparisons (Figs. Figures 6, 7 and 8), we settled on the use of 1-day relaxation time-scale as the model physics performance (visually) was better when using a short time-scale. Following up on this decision, we hypothesized that the physical correction will also lead to improved performance of the modelled biogeochemistry. It is important to objectively analyse and quantify the improvements and prove that the hypothesis is valid.

The bias and rmse across the different BGC-Argo tracks used in this study for each relaxation time-scale is depicted in Figs. 3, 6, 7 and 8 (Fig. 13). Since the effect of the relaxation time-scales were more prominent during the winter/spring periods (Fig. 7 and 8), and since the days are short in the high latitudes during winter, the statistics we present in Fig. 13 are the averages of the data points of Fig. 12 for March - May for 0 - 200 m depth range. In general, Figure 13 depicts a common pattern where as the relaxation time-scale increases, the simulations are statistically less representative of the observed data, with some BGC-Argo track experiments statistics vary more than others. The 6902547 case which we present in detail in this study varies moderately. Since both the physics and the chlorophyll *a* respond to relaxation time-scale more positively with

shorter time-scales, for this kind of approach, we suggest using as short time-scale as possible. We note that shorter than 10-days relaxation scale, the differences among the statistics are marginal. The choice of the scale will in these cases be a decision on how flexible towards the BGC-Argo T and S the model physics are intended. The model MLD for 10-days scale (Fig. 8) is comparable to that of the 1-day scale.

One can argue that the statistical differences between the time-scale experiments are not strikingly different, and, as such, the choice of a different time-scale would be arbitrary. We argue that even if the statistical differences are relatively low for the particular case of the Nordic Seas, the relaxation to climatology (WOA18) represents statistically the least capable. Using climatology data, the relaxation scheme will miss anomaly events and will thus poorly represent inter-annual variability. Capturing such variability plays an important role in understanding the ecosytem dynamics of regions where the seasonal and annual productivity is highly dependent on the strength and duration of vertical mixing events due to nutrient limitation such as the subtropical North Atlantic (Siegel et al., 1999; Neuer et al., 2007; Helmke et al., 2010; Yumruktepe et al., 2020). Correcting the model winter physics (e.g. timing and depth extent of MLD and mesoscale activity) will not only improve the model biogeochemistry in general, but will introduce inter-annual variability.

The framework we present in this study may yield important applications for regions of nutrient limitation. In our case, except the summer months, the biogeochemistry is not nutrient limited (Figure 10b-d), and the correction we applied using the BGC-Argo physics therfore mainly related to the light-limitation and the strength of vertical mixing. While improving the strength of convective mixing, we improved the timing of the spring bloom but the overall effect to the total productivity would possibly be more significant in nutrient-limited regions. This hypothesis was not tested in this study, but we suggest this as a valuable follow-up study.

## 4 Concluding remarks

In this study, we established a 1D ocean modelling framework that employs BGC-Argo buoys for improving and evaluating the biogeochemical process representations. The code we developed for this study is written in a generic format, and any BGC-Argo (or Argo in case of biogeochemical validation is not required) trajectory data can be used as the physical setting for a biogeochemistry simulation. The supplied code (see Code and data availability), prepares a 1D model setup by creating atmospheric forcing, climatological and Argo relaxation data along the Argo track. The framework uses the well established GOTM physics model and FABM biogeochemical coupler. These allow for a quick setup of the experiments and application of a wide-range of biogeochemical models. In this study we focused on presenting the framework, its approach and an example use case. The use case presents how the modeller can setup the experiments from scratch, thereby this article is also a guide for replicating similar experiments with any BGC-Argo buoy.

In addition to the presentation of technical details, we showcase how the framework can be used to improve biogeochemical models using both the physical and biogeochemical Argo data. As our focus on this study was to present the technical approach, our experiments are limited to simulating the observed chlorophyll *a* from a limited number of buoys. Thus an immediate follow-up work for this study is essential to utilize the framework towards a scientific focused approach. We suggest extending

the approach using other BGC-Argo variables such as oxygen, light, nitrate and particle backscatter. A combined use of these variables in the analyses would provide a more complete picture on light vs nutrient limitation, timing and depth of biogeochemical processes and an estimate on organic matter concentrations. Such an assessment would allow for a more in depth tuning of model dynamics. An interesting approach would be to construct a similar study at regions where the oceanic

production is highly dependent on vertical transfer of nutrients to the surface, as such, improving the model physics using the BGC-Argo buoys may benefit the modelling studies at regions of high nutrient-limitation. As the BGC-Argo dataset by now has matured over a decade, together with the improved models, a regional (or global) assessment of the biogeochemical variables may allow an understanding on the trends of these variables.

On the topic of tuning, a planned follow-up to this study is the application of more systematic parameter tuning approaches

(e.g., Gharamti et al., 2017) compared to the relatively simple exercise presented here. In the future, the number of buoys that are used in the experiments should be increased, preferably establishing a region-wide sample set, and a detailed sensitivity analysis should be made for a wider parameter set. For instance, we note that the model chlorophyll *a* is on the higher end with the parameter values chosen in this study (see Section 3.3.3). Dedicated parameter tuning approaches should consider uncertainties in the BGC-Argo data as well as the uncertainty range of the tuned parameters. Because this study focus on

improving the model formulation (i.e. increasing growth rates under low-light conditions), rather than model parameter fine tuning, a dedicated assessment of BGC-Argo data errors was not included. To limit the effects of observation uncertainties, we only included (see Sec. 2.1.1) the "ADJUSTED" BGC-Argo variables (i.e. temperature, salinity, pressure and chlorophyll *a*) which provide either a "real-time-adjusted" or a "delayed-mode" data control and correction (Bittig et al., 2019). Despite applying a level of correction, there are still observational errors present. For example, the measured fluorescence chlorophyll

*a* to chlorophyll *a* ratio can vary due to various factors and can lead to uncertainty as high as $\pm 300\%$ (Roesler et al., 2017). However, some of these errors can be reduced to a maximum of $0.12 \, \mathrm{mg \, Chl \, m^{-3}}$, with an average reduction of $\pm 40\%$ (Johnson et al., 2017; Bittig et al., 2019). On the other hand, the BGC-Argo estimated POC uncertainty is lower and but can be as high as $40 \, \mathrm{mg \, C \, m^{-3}}$, about $50\%$. In the case of oxygen, the sensors show a strong drift (order $-5\% \, \mathrm{year^{-1}}$ between calibration and deployment), this can be corrected (to approx. $1.0 - 1.5 \, \mu\mathrm{mol \, kg^{-1}}$) with surface measurements adjustments along-the-track

(Bittig et al., 2018). Similar uncertainties that exist for all other BGC-Argo variables should also be accounted for in model validation studies.

We envision an ensemble simulation approach for model biogeochemical parameter tuning as a follow-up study where we construct a suite of ensemble experiments with systematic perturbations of selected model parameters within a $\pm$ uncertainty range from the respective reference parameter value. However, depending on the number of modified parameters and BGC-

Argos, the number of experiment can be in the range of thousands which raises the question of how to select the parameter set(s) that yield the best results objectively. The statistical analyses that have been performed in this study is done on a limited number of BGC-Argos and a single biogeochemical variable (i.e. chlorophyll *a*) and may turn out inconclusive for a fine-tuning parameter study, given the BGC-Argo uncertainty. Newer BGC-Argos are equipped with multiple biogeochemical sensors, making the statistical analysis of a parameter fine-tuning experiment more robust as the number of experiments increases

while accounting for multiple BGC-Argo variables including their associated uncertainty ranges. The inclusion of multiple

BGC-Argo variables statistics would enhance the ecosystem representation of the parameters, and multiple variables would provide more constraints toward realistic representations. At that stage of the analyses, the uncertainty range of the observed variables can be included, and the search for the better performing experiments could be narrowed down for the less uncertain variables (e.g. POC, oxygen) and widened for the more uncertain ones (e.g. chlorophyll *a*). In addition, instead of directly incorporating the concentrations of the full experiment in the statistical analyses, it may provide valuable insights to separately assess the timing of seasonal events driven by the mixed layer dynamics. Alternatively, comparing correlations between the model and the depth location of key features in the BGC-Argo profiles, such as the nutricline, would give an insight to the mixing and production dynamics (e.g., Salon et al., 2019). These approaches would reduce the influence of observation errors, but would rely on the consistency of the sensor along-the-track. Finally, a more elaborate data assimilation scheme that take into account model variable and parameter uncertainties, such as that based on Ensemble Kalman Filter could be considered to this framework in an idealized setting to investigate whether or not the current model parameterization is suitable to represent the observed real world process (e.g., Singh et al., 2022).

In parallel to fine-tuning the model parameters, it is equally important to further evaluate mechanistic approaches to phytoplankton growth, mortality, grazing pressure, and organic matter export dynamics. Such mechanistic approaches allow the models to adapt to changing environmental conditions both for regional coverage, changes in climate and the state of the oceans. Our study demonstrate the possibility to the design and apply such approaches through considering different (1) regional coverage and (2) ever-growing time-extent of BGC-Argo data, allowing us to investigate the model discrepancy on a large-scale but also on local scale when considering (3) high-resolution depth and time coverage. The study presented here is an example of the latter where we detected a shortcoming of the simulated primary production in low-light environments, and applied a mechanistic change to the model chlorophyll *a* dynamics with minor parameter tuning. This application was an initial attempt to showcase the capacity of the framework for the wider scientific community and a follow-up study that focuses on further mechanistic approaches is a natural extension of this study.

With this framework, we provide modellers with an alternative/additional dataset to in situ or remote sensing data, cover wider regions, depths and time-periods, and with least effort, make these data points available for model evaluation. We vision advancements in the understanding of the functioning of marine biogeochemistry through the use of models, which will be improved with the use of this framework. The cost effective nature of this framework should allow (1) the employment of multiple models of variable complexity, (2) application of them to various regions and time-periods to cover multiple ecosystems, (3) improve and fine tune process formulations, (4) compare the results for a beyond model-specific approach, (5) ultimately, and most importantly, apply these models in a 3D setting covering a range of use cases from operational oceanography (e.g., Yumruktepe et al., 2022b), to regional/global links to higher trophic levels (e.g., Utne et al., 2012) or to climate (e.g., Tjiputra et al., 2020). Through these use cases, models have the potential to be an important ally to the observations we have for an improved understanding of nature and the future of our oceans.

*Code and data availability.* The exact version of ECOSMO II(CHL) model used to produce the results used in this paper is archived on Zenodo (Yumruktepe et al., 2022a, https://doi.org/10.5281/zenodo.7773509) including the input data and scripts to produce model input, run

the model and produce plots for all the simulations presented in this paper. They are openly available under a Creative Commons Attribution 4.0 International license. The GOTM model and FABM coupler are developed at https://github.com/fabm-model/fabm/wiki/GOTM (last access: 24 February 2023) and the version used (GOTM version 7 revision: 2219) is included at Yumruktepe et al. (2022a) under 'gotm' directory. Yumruktepe et al. (2022a) includes a shell script, 'build_gotm_fabm.sh', that will by default install the GOTM-FABM version we have used in this study. The ECOSMO II(CHL) model code is written in FORTRAN. Pre- and post-processing scripts are written in Python

3. Yumruktepe et al. (2022a) includes various folders. The experiment we presented here is stored in 'experiments' folder and can be used as an example for other experiments. 'figure_codes' folder includes the scripts to reproduce the figures in this paper. 'nersc' folder includes ECOSMO II(CHL) code to be coupled with FABM during compilation. 'build_gotm_fabm.sh' builds the coupled GOTM-FABM-ECOSMO II(CHL). Model outputs and data used for figures are stored in 'output_files'. In the main folder of Yumruktepe et al. (2022a), 'argoinput.py' is located. This is a compilation of the Python subroutines that are used to build and post-process the experiments. It is recommended to

either copy this script to your Python path or use a symbolic link where the python scripts are executed. Investigate 'README.md' for further instructions.

## Appendix A

### A1  BGC-Argo 6902547 modelled MLDs

This section includes the extended time-period and the full set of simulations depicted in Figure A1. Due to the inclusion of

590 multiple years and 12 different time-series data presents a challenging readability of the figure. However, inclusion of each data-set in the figure is essential for the understanding of the model dynamics as a response to different relaxation time-scales. Therefore, we included the simpler form in the main text (Fig. 8) and the full set here in the Appendix.

### A2  Observational data sources

The input files that were used for the discussed experiment in the study are archived on Zenodo at Yumruktepe et al. (2022a)

595 'experiments/example' folder. This folder includes the necessary files to reproduce the experiment. The following web links are the sources of these input files and 'build_experiment.sh' included at Yumruktepe et al. (2022a) is configured to connect to these links for the preparation of a new experiment.

1. World Ocean Atlas 2018, temperature:
   https://www.ncei.noaa.gov/access/world-ocean-atlas-2018/bin/woa18.pl?parameter=t.
(last access: 07 February 2023)

2. World Ocean Atlas 2018, salinity:
   https://www.ncei.noaa.gov/access/world-ocean-atlas-2018/bin/woa18.pl?parameter=s.
   (last access: 07 February 2023)

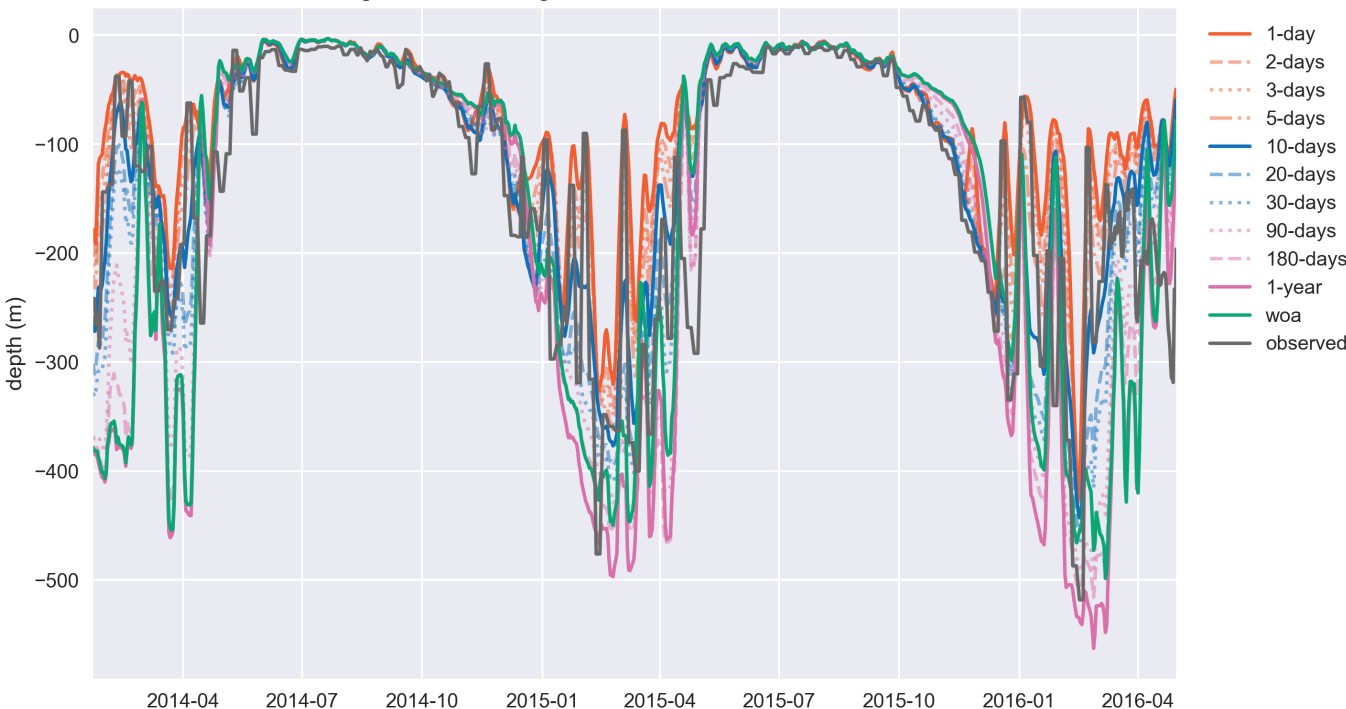

**Figure A1.** MLD, estimated by using 0.03 kg m$^{-3}$ density change criteria from 10 meters, similar to the cases of temperature and salinity, is represented better in experiments that use shorter relaxation time scales. Model MLD depicted in this figure is GOTM model output 'MLD_surf' calculated from turbulence. This figure depicts the full time period of the BGC-Argo track and model experiments. The simulation identified as 'woa' corresponds to REF-WOA-1year and the remaining simulations correspond to REF-Argo- with different time-scales of relaxation.

3. World Ocean Atlas 2018, nitrate:

    https://www.ncei.noaa.gov/access/world-ocean-atlas-2018/bin/woa18oxnu.pl?parameter=n.

    (last access: 07 February 2023)

4. World Ocean Atlas 2018, silicate:

    https://www.ncei.noaa.gov/access/world-ocean-atlas-2018/bin/woa18oxnu.pl?parameter=i.

    (last access: 07 February 2023)

5. World Ocean Atlas 2018, phosphate:

    https://www.ncei.noaa.gov/access/world-ocean-atlas-2018/bin/woa18oxnu.pl?parameter=p.

    (last access: 07 February 2023)

6. Ocean colour climate change initiative v5.0, chlorophyll *a* and kd$_{490}$:

   https://rsg.pml.ac.uk/thredds/catalog/cci/v5.0-release/geographic/daily/catalog.html.

(last access: 07 February 2023)

7. BGC-Argo along-track data:

   https://data.marine.copernicus.eu/product/INSITU_GLO_BGC_DISCRETE_MY_013_046/description.

   (last access: 07 February 2023)

8. ERA5 atmospheric forcing data:

https://cds.climate.copernicus.eu/cdsapp#!/dataset/reanalysis-era5-single-levels?tab=form.

   (last access: 07 February 2023)

*Author contributions.* VÇY built the framework and together with EAM, JT and AS, they designed the experiments and VÇY carried them out. VÇY and AS are the developers of ECOSMO II(CHL). EAM and JT coupled their respective biogeochemical models to FABM (not presented here) for a follow-up work. Their experience with the coupling helped generalizing the framework code for a wider community.
VÇY prepared the paper with contributions from all co-authors.

*Competing interests.* The authors declare that they have no conflict of interest.

*Acknowledgements.* The authors acknowledge the support of the Bjerknes Centre for Climate Research, Norway. Model input data was stored and used through the Norwegian Sigma2 infrastructure under the project NS9481K.

*Financial support.* This research was financially supported by the Bjerknes Centre for Climate Research projects Fast Track Initiative and the The Breathing Ocean.

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
