# Peer review of "An along-track biogeochemical Argo modelling framework, a case study of model improvements for the Nordic Seas"

_Geoscientific Model Development, 2023_

## Author Comment (AC1)

We thank the reviewers for their comments. We report here our responses to the specific points.

We start with our responses to RC1, and follow it by RC2. The text in *blue italics* are the comments from the referees. The text in "black regular" is our response to the comment, and the text in *red italics* is our text that appears in the revised manuscript.

We also would like to thank the comment of CC1 from Martí Galí. We included a response following those of the referees.

We thank all reviewers for their constructive feedback. They provided very valuable insight to what we are trying to achieve with this study.

Comments from RC1:

*The authors present a study that use BGC Argo data and model experiments in tandem to drive and evaluate the model. Argo data are used to both help simulate a realistic physical environment and interpret biogeochemistry. The framework presented is intended to have minimal technical effort and set up ideas for future synthetic modeling activities in the future as the Argo fleet grows. The article focuses on chlorophyll, which is known to be highly variable and not the best metric for biomass. The model itself is not as simple as advertised, as it includes assumptions about the partitioning of biomass into different groups and many more parameters than in the simplest possible case if one if mainly interested in phytoplankton stocks or physiology (which seems like the goal, given that chl is emphasized). Other high quality metrics are available, which could allow model simplification and interpretation beyond what was considered herein. Argo floats also have many more biogeochemical variates that are not well ultilized in this study, including oxygen (which can be used to assess zooplankton in some capacity) and bbp (which can be used to assess accumulation rates directly rather than through a model). I suggest either simplifying the model so that improved interpreation can be done, or adding complexity to the analysis to better constrain the model in its current construction (ie, incorporating Argo nutrients and metrics for zooplankton). More detailed comments follow.*

We emphasize that our main objective with this study is to present the framework we have built that uses BGC-Argo data for model assessment and improvement. The paper is written in a way to present the technical details of the framework and showcase its capacity (and simplicity) as a model assessment tool which can be a valuable alternative to other observational data such as in situ ship samples or satellite images , particularly since BGC-Argo data provide multi-variable data that is both vertically and temporally high resolution. In addition to showcasing the framework application, using the ECOSMO II(CHL) model as an example, we also used the framework to detect shortcomings of the model, applied a minor change to its formulation and tuned its parameterization. We finally assessed the changes using BGC-Argo chl-a data.

Referencing the application of ECOSMO case, we advertised the framework to be a simple tool, which we believe it is. We accept that the codes provided may be overwhelming as they are handling data from multiple sources, filtering and interpolating (and much more),

but as a tool, it only requires a few lines of entry from the user's side to compile the model and build an experiment along any BGC-Argo track. The statistical analysis codes are built into the master code, and the description on the use of these codes are provided. However, the model that is ECOSMO itself is not advertised as simple, in fact it is described as intermediate-complexity lower trophic level marine ecosystem model both in this manuscript (line 103) and the ECOSMO reference paper (Yumruktepe et al., 2022). But the complexity of ECOSMO, we believe, is not relevant to the manuscript, as it is one of the many models that is coupled to FABM coupler, thus from the point of view of the framework, the choice of model is generic. In fact, when the referee suggests simplifying ECOSMO, they point out a very important strong side of our framework. Any biogeochemical model can be easily used with it ranging from a simple NPZD type model to a very complex one sharing the same physical setup (e.g. along the BGC-Argo tracks we provided) allowing a cross-model validation, and this ensemble of models will, in addition to the simple models as the referee suggests, provide a valuable insight to the biogeochemistry of the modelled region. The referee also suggests using the alternative BGC-Argo variables (e.g. bbp, oxygen, we would like to add nitrate and light to this list), and that is precisely what we want to do as a follow-up to this study. A full-scale model parameter tuning, increasing or simplifying model complexity, or a model assessment using multiple BGC-Argo variables would be too out of scope of this manuscript and would be very overwhelming for the reader, but noting that, as we mentioned, it is a natural way to proceed. And in fact, we are working on a follow-up work using this framework, employing an additional biogeochemical model, and including oxygen. Our goal is to focus on the technical details of the framework itself, and showcasing how it is used, such that the paper will serve as a reference point for many follow-up studies.

We understand the concern on the use of chl-a, and the referee has a very valid point. But we would like to note that the manuscript is to provide an example approach. We chose chl-a because we are actively using in situ and satellite chl-a from other sources to assess ECOSMO in our operational system, we apply a simple near-real time chl-a data assimilation to the model surface chl-a and constructing a vertical profile using satellite data. The choice of chl-a was mainly for practical reasons as we are actively using chl-a with ECOSMO assessment and operational modelling studies in Copernicus marine services (Arctic Ocean Biogeochemistry Analysis and Forecast | Copernicus Marine MyOcean Viewer), and we have a roadmap towards the use of this variable. If the Quality information document in the link is inspected, ECOSMO is highly invested in validation using chl-a, and it was natural for us to use chl-a here as well as a starting point. In the follow-up studies, other variables will be equally important such as oxygen and bbp as the referee suggests. We have plans to assess model POC, and nitrate when we are confident in using them.

In summary, we hope that the primary intention of this study is better explained. We understand that some points are overlooked, and in the revision, the points the referee raises and our responses will be discussed in detail, and statements will be place where relevant in the text.

As the following comments are in line with the general introduction on the referee, we refer to those comments for our responses and the added text. However, our modified paragraph to the introduction can be referenced here. We elaborate more on our aims and objectives.

Here is the added/modified text:

(lines 46 - 57):

*In this study, we focus on using BGC-Argo as an additional observational data source to in situ sampling and remote sensing, and how to take advantage of two important aspects of the BGC-Argo dataset: (1) its regional and temporal coverage, (2) combined availability of high-resolution physical and biogeochemical data. Our main objective is to establish the framework and showcase its capacity as a tool for model development and assessment. The framework will allow the modeller to construct a Lagrangian type experiment along a BGC-Argo track in order to visually and objectively assesses the model performance and subsequently advance its dynamics and optimise its parameters. Even though one of the ultimate aims of using this framework for a modelling study is the assessment of the observed biogeochemistry, our primary aim is to present the details of the framework. Therefore a full assessment of the observed biogeochemical variables is outside the scope of this study. Here, we present how BGC-Argo physical data can enhance the realism of model physics, thereby allowing the evaluation and improvement of the modeled biogeochemistry. Specifically, we show how its high-resolution vertical and temporal chlorophyll a sampling can be used to advance model formulation, and objectively assess the model parameters. The ultimate goal is to establish a 1D modelling framework towards improving regional and global models.*

*Satellite merged products have known issues (see van Oostende et al 2022). It may be better to have a consistent mission value to evaluate the BGC-Argo data, as OC CCI is essentially a modeled product.*

We thank the referee for this valuable comment. After inspecting van Oostende et al. (2022), we see in Figure 1 that the time period May 2012 – May 2016 is free of sudden steps or inconsistencies, where the time-frame fits our model setup period. For this reason, we revised Figure 4 (line 268) to this period only. Our objective with this section is not to validate BGC-Argo chl-a as it is extensively covered by the product producers, but show the reader how it performs for the Arctic specific case.

*Here is the revised figure:*

[Figure]

*Figure 4. Chlorophyll a statistical analyses of in situ bottle samples with a search radius of (a) 2 km, (c) 10 km and BGC-Argo with a search radius of (b) 2 km, (d) 10 km reveal that BGC-Argo statistics against satellite chlorophyll a show the same pattern as in situ bottle statistics against satellite chlorophyll a. The computed statistics and number of sample points for each sample set is depicted in the figures. Equations for the computed statistics are described in Sec. 2.2. Data from all sources are log10 transformed.*

Here is the revised text:
(line 247-251):

*We note that van Oostende et al. (2022) shows inconsistencies within the continuity of OC CCI v5.0 chlorophyll a product appearing as sudden steps in the time-series. These steps appear when a satellite is launched or removed. For this reason, we limited our statistical analysis from May 2012 to May 2016 where only MODIS and VIIRS are continuously active, and the time frame fits our study period. Figure 1 in van Oostende et al. (2022) depicts no sudden steps in the OC CCI V5.0 data for that period.*

*Section 2.2 How many matchups did you obtain? What was the time separation between samples? 2km may be good for +/-a few hours of a matchup, but if more time is used, a great physical distance should be used.*

We have used daily satellite data in our analyses. A note on the number of matchups is added to Figure 4. We also evaluated the matchup statistics for a greater search radius, i.e. 10 km. A second row is added to Figure 4, thus giving the reader a comparison of how the matchups look within a range of distances.

*Line 140: What about parameterizing C:Chl variability with temperature or nutrient stress as well?*

Mechanistic additions to model dynamics is important as such the model is expected to be more suitable to changing environmental conditions, in this case, the referee suggest temperature and nutrient stress which are important. In reference to our point above for model code variations, we would like to address this valid point in the follow-up studies. We added a dedicated paragraph on the possible changes to model structure.

The following is the added/modified text:
(lines 519-528):

*In parallel to fine-tuning the model parameters, it is equally important to evaluate further mechanistic approaches to phytoplankton growth and mortality, grazing pressure, and organic matter export dynamics. Such mechanistic approaches allow the models to adapt to changing environmental conditions both for regional coverage, changes in climate and the state of the oceans. The study we present allows the designing and application of such approaches through (1) regional coverage and (2) ever-growing time-extent of BGC-Argo data allowing the investigation of model discrepancy on a large-scale to (3) high-resolution depth and time coverage allowing the evaluation of the model at a local-scale. The study presented here is an example of the latter where we detected a shortcoming of the model production in low-light environments, and applied a mechanistic change to the model chlorophyll a dynamics with minor parameter tuning. This application was an initial attempt to showcase the capacity of the framework for the wider scientific community and a follow-up study that focuses on further mechanistic approaches is a natural extension of this study.*

*Table 2: Why is grazing rate held constant rather than fluctuating with standing stock of phytoplankton? Should the table be revised to say 'max grazing rate?'*

The referee is correct. The rate given in the table is the maximum grazing rate. The resulting grazing rate based on food preference and availability is calculated by Eq 15. In the revised version, we include the symbols to the parameters provided in the table, so that the reader can easily track which parameter we are referring to among the Equations. (line 149)

*Table 2: shouldn't mortality rate be a function of growth rate or concentration (viruses) than a fixed number held constant with any concentration?*

It is a valid point and we may consider this in the future iterations of ECOSMO as we note in the manuscript that the model is on the productive side compared to the observed data, and this addition may suppress excessive growth during already highly-productive seasons. Please refer to our comment above in response to *'What about parameterizing C:Chl variability …'*

*Line 150-155. The model has so many parameters that many can be tuned in different combinations to match the observations. Given that a goal is biogeochemical interpretation, how can the number of free parameters be justified? A simpler model may be a better place to start before adding phytoplankton and zooplankton groups.*

We emphasize that our primary goal with this work is to present the framework for model evaluation and development. We do not envision biogeochemical interpretation for this particular study. However, we are committed to the long-term development of this framework and in the follow-up studies, we will definitely target biogeochemical interpretations. Please see our response to comment (starting with *Finally, I'm curious …*) of the same referee below.

*Line 160: What resolution is required specifically for temporal variations?*
We aim to capture important developments in plankton biomass, for example the timing and duration of the spring bloom. This is important in order to be able to comment on the physical-biogeochemical interactions, and thus incorporate model solutions to it. We added a text (lines XXX) on this in the revision and cited Silva et al. (2021) whom depicted the timing and duration of the spring bloom in the Nordic Seas, where they state a duration of 28 – 58 days, which is a much narrower time frame for an on-board sampling. Argos provide 5-10 days typically, which is enough to capture the onset, peak and decay of the spring bloom.

The following is the added/modified text:
(lines 164-168):
*In the case of temporal resolution, Silva et al. (2021) gives a range of 28 - 58 days for the duration of the spring bloom for the Norwegian and Barents Seas. It is highly unlikely that a conventional on-board in situ observations would provide the samples to cover the onset, peak and decay of the spring bloom within a large regional area, whereas with sampling frequency of 5 - 10 day, BGC-Argos can capture the changes for the duration of the spring bloom in the Nordic Seas.*

*Line 163: What is meant by 'relatively close' in a quantitative sense?*
Changed the wording to "sampling a continuous and similar water mass". (line 171)

*Section 2.4.2 How are uncertainties in Argo values incorporated into the model (for example, chl, which even when corrected, can have errors, e.g., your figure 4? RMSE of ~ 0.27 or 0.29 in log10 space or by itself (not clear from the figure legend if log10 was applied to obs) is nontrivial)*
We added a note on the figure caption that all the data are in log10 scale.

We also added a paragraph discussing the relation between the Argo uncertainty and model changes. In summary, even though Argo has mismatches with the true values, they present valuable information to improve the model physics, detect fundamental differences in biogeochemistry such as timing and strength of the spring bloom, or the absence of the deep chlorophyl-a maximum. BGC-Argo may not be precise with the exact value of chl-a, but will point out the presence of biomass which sometimes models are not able to represent. Based on this, we applied a mechanistic change to the model code to reflect on the absence of these fundamental issues, thereby improving the model representation of these cases. We are not pursuing the representation of precise concentration values.

The following is the added/modified text:
(lines 348-361):
*Prior to discussing the changes to the model, it is important to elaborate on the effect of uncertainty of the BGC-Argo data, for we rely on this dataset to exert changes to the model code and parameterization. As is the nature of observations, they all are different than the true value, and there will be mismatches (Skogen et al., 2021), even in the case of in situ chlorophyll a bottle samples. Nevertheless, while we acknowledge that there are mismatches among different datasets (Fig. 4), we can still retrieve enough information from the BGC-Argo dataset to detect model shortcomings and propose improvements. For example, in every case where the model was nudged towards the BGC-Argo temperature, stronger relaxations result in a better match between the model T and SST which is an independent dataset to BGC-Argo (Fig. 7). Similarly, in the case of BGC-Argo chlorophyll a uncertainty, we are not pursuing a precise 1-to-1 match between the model and BGC-Argo, but exploring notable differences that should be improved regardless of the concentration differences. As such, there are fundamental errors in the model that need to be addressed, i.e. the late-bloom which disrupts the timing of energy transfer to the upper trophic levels and the absence of DCM which is the production that is not accounted for in the model. These fundamental dynamics are observed in the BGC-Argo data even if they may not be represented by precise accuracy. Therefore, in the experimental phase, we focus on these two issues and investigate ways to improve the mechanics of the model in general. Noting these, fine-tuning model parameters in a follow-up study would require a more research on the effect of BGC-Argo data uncertainty.*

*Why is chlorophyll used from BGC Argo rather than bbp, which is shown to be more reliable with satellite data and also with phytoplankton biomass? Using bbp allows one to calculate both standing stocks (Graff et al 2015) and accumulation rates, which may be compared to the model and allow physiological model errors (Chl:C ratios) to be irrelevant. I know that the authors list bbp and other Argo variates in the concluding remarks. However, bbp may be a simpler case study for the authors to examine. The other Argo variates could be used or discussed for creative model development. For example, zooplankton can be parameterized from Argo data and that is not mentioned well in the text.*
The referee has a valid point. While we have plans to incorporate bbp in the follow-up studies, including nitrate, oxygen and light, the choice of chl-a for this study was a practical one. Our attempt is to showcase the framework and because ECOSMO is incorporated to Copernicus marine systems operational and reanalysis model validation and assimilation frameworks, we have substantial practice on the use and interpretation of chl-a data. Therefore, using chl-a as an example proved to be the preferred choice and we were able to focus on the technical details of the work. We have plans to use bbp as a means to estimate POC (and export) but as the referee suggests, it may ultimately replace chl-a.

*Finally, I'm curious about the choice of the model. I'm not surprised that the model can be tuned to match the obserations because it has so many free parameters. The goal is eventually moving beyond an accurate deterministic model and being able to interpret and attribute changes to biogeochemical function. How can that be accomplished given the*

*number of assumptions listed herein? More text on that point will help the reader hoping to employ a similar analysis. I'm not clear on how the model can be used to extend interpretation beyond what can be done from the Argo observations alone.*

As stated above, our primary goal with this work is to present the framework for model evaluation and development. We envision this work as the reference study for other modelers to use for their respective models. Our ultimate goal, on the other hand, is to allow the use of multiple models, improve their model configurations for an improved representations in 3D. Improved 3D configurations will link to regional/global models with various focuses. That is how we try to go beyond the accurate deterministic model configuration. It is a long-term objective, and this paper is a start. As a first step, our aim here, just like other conventional observational data, such as satellite or in situ bottle data, is to create a simple way to construct a base for model-data comparison. In parallel to satellite data, with Argo, we cover below the surface, or compared in situ data, we cover wider regions at a high resolution temporal scale. While we are constructing experiments along the tracks, in core, we are establishing model evaluations, but with a higher number of data points, which may not be as technically easy for many to acquire. We also provide a framework that makes it easy to employ a range of biogeochemical models seamlessly through the FABM coupler. With that, we have the opportunity to evaluate models of different complexity, just like the referee suggests, a simple one to a very complex one. A cross comparison on these in a Lagrangian fashion would provide valuable information on the use of models, and improvements in general when applied to 3D would improve our understanding. Even in the case of climate models, their core biogeochemical models can easily be incorporated here, and their shortcomings tested which would be highly expensive in 3D at a climate time-frame. Please see our additions to the concluding remarks section.

The following is the added/modified text:
(lines 529-539)
*With this framework, we provide modellers with an alternative/additional dataset to in situ or remote sensing data, cover wider regions, depths and time-periods, and with least effort, make these data points available for model evaluation. We vision advancements in the understanding of the functioning of marine biogeochemistry through the use of models, which will be improved with the use of this framework. The cost effective nature of this framework should allow (1) the employment of multiple models of variable complexity, (2) application of them to various regions and time-periods to cover multiple ecosystems, (3) improve and fine tune process formulations, (4) compare the results for a beyond model-specific approach, (5) ultimately, and most importantly, apply these models in a 3D setting covering a range of use cases from operational oceanography \citep[e.g.,][]{ecosmo2chl}, to regional/global links to higher trophic levels (e.g., Yumruktepe et al., 2022b), to regional/global links to higher trophic levels (e.g., Utne et al., 2012) or to climate (e.g., Tjiputra et al., 2020).. Through these use cases, models have the potential to be an important ally to the observations we have for an improved understanding of nature and the future of our oceans.*

Comments from RC2:

*This paper presents a designed framework linking Biogeochemical-Argo (BGC-Argo) observations to biogeochemical models in the Nordic Seas. The BGC-Argo and satellite surface temperature were used to evaluate the simulated temperature, salinity, and mixed layer depth. The Modeled chlorophyll a (chl-a) was evaluated/compared against the BGC-Argo chl-a along the BGC-Argo trajectory. The differences between modelled chl-a and BGC-Argo chl-a (Figure 9) indicated that (1) the model failed to reproduce the deep chlorophyll maxima throughout June to September in the 20-50m depth range, and (2) the timing of spring bloom initiation is late from the model. To address these differences, the authors tried*

*to improve the model, but in phytoplankton growth formulation and parameterization section, there are lots of hypotheses, will the hypotheses affect the model setup and outputs?*

As with all models, the underlying hypotheses and assumptions will affect the model setup and output. A major reason for comparing to observational data is to assess if major flaws are present in this theoretical founding. In our case, the timing of the spring bloom and the subsequent deep production was not well represented, so a change of the formulation was deemed necessary. In this case, it was hypothesized that including a variable chlorophyll a to carbon content in the light limitation term would be a plausible improvement, as such the model has improved response to light in low-light conditions. The subsequent assessment of the model statistics supported that this was case and gives weight to the importance of this mechanism. But we emphasize that the model used in this paper is just an example of how the framework can be used. A whole range of hypotheses could be stated and tested against each other. So yes, the hypothesis will affect the model setup and output, but the assessment framework is independent from the model setup and can be used to test hypotheses, model tuning and as well validation.

RC1 also commented on various hypotheses. Please see our response and refer to the related text. The referee can follow the comments to *Line 140: What about parameterizing C:Chl variability with temperature or nutrient stress as well?* and *Table 2: shouldn't mortality rate be a function of growth rate or concentration (viruses) than a fixed number held constant with any concentration?* And finally, our response to the comment *Finally, I'm curious about the choice of the model …* is a good example how we envision the use of this framework and hypotheses within.

*It is hard to see which result is better when compared Fig. 11c (chl-a difference between improved model and BGC-Argo) with Fig. 9e (chl-a difference between model and BGC-Argo).*

Figure 12 provides a statistical assessment of the improvements going from the reference model to the improved model. This assessment clearly shows a reduction in both bias and root mean square error, especially relating to the DCM, supporting the visual assessment. While a direct visual assessment can be difficult, we still think it is warranted to describe.

*Also, in the paper, I did not see the discussion on uncertainty of BGC-Argo observations, such as what's the uncertainty of chlorophyll a, temperature, and salinity from BGC-Argo? Will the uncertainty affect the comparisons between the modelled results and BGC-Argo results?*

In general, the uncertainty and representability of observations used to assess models are important. Models that are tuned and/or validated against uncertain observations potentially inherit that uncertainty. This is a general challenge in ecological modeling as observations are often sparse and unevenly sampled in space and time (Skogen et al., 2021 https://doi.org/10.3354/meps13574). Nonetheless, some degree of ground-truthing is necessary even if it is based on incomplete and uncertain observational data. Using BGC-Argo data is, in principle, subject to the same challenges.

However, we argue that the approach described in our paper goes a long way to improve on existing methods of comparison. Firstly, BGC-Argo data was subset to exclude data points with low quality. Secondly, an assessment of the BGC-Argo chlorophyll a was made, comparing it to values from satellite derived chlorophyll a (please see the changes to Fig. 4 in reference to RC1 comments starting with *Satellite merged products have known issues* and *Section 2.2 How many matchups did you obtain?*). Thirdly, while the paper focuses on one BGC-Argo floats, multiple floats were tested and we discuss the importance of using multiple floats in the assessment (Lines 502-519). In extension, if the BGC model is to be used in a 3D setup, additional validation is necessary.

Finally, and most importantly, the framework describes a method to reduce the problem of incompatibility. BGC models are closely coupled to the modelled physics, so when the output is compared to observations, it is challenging to assess if any discrepancies between the model and the observations are caused by differences in the physical environment or by a wrong implementation/parameterization of the biological processes. The approach described relaxes the modelled physical environment towards the observed, thus allowing the biological part to be targeted.

A full assessment of BGC-Argo uncertainties is, in our opinion, beyond the scope of this paper. However, we recognize that observation uncertainty and representation is important, and have included a paragraph. RC1 also raised this issue. We refer to that comment starting with *Section 2.4.2 How are uncertainties in Argo values incorporated into the model …*

Comments from CC1:

The article of Yumruktepe and colleagues presents a new approach that allows using BGC-Argo data to improve biogeochemical models. By relaxing the 1D model physics towards Argo float observations, the authors can subsequently focus on improving the biogeochemical model. I read with great interest the article and I believe this approach represents a step forward in this developing field of research.

As a general note, one potential flaw of "along-track" approaches, whereby 1D models are matched to observations made by individual Argo floats, is the assumption of float Lagrangianity. Given that floats profile between 1000 m (sometimes 2000 m) and the surface, it is unlikely that they can track the same water masses at all depths over an extended period. Therefore, very strong relaxation towards observations may force the 1D model physics beyond what is physically reasonable.

We agree with the comment. In our tests, considering that we had access to 50+ BGC-Argo floats we only settled on <10 of them. We applied, what we believe a strict filtering process. We only included floats with "_ADJUSTED" variables, applied flags to keep good and adjusted data. We filtered and kept BGC-Argos that would remain in similar watermass. Finally chose floats that would complete at least a year of trajectory, but would not provide a too long in time data. The reason is that, with long term simulations with the absence of lateral interactions (1D setup), you can conserve biogeochemical conditions with nutrient relaxations, which we turned off to focus on the processes themselves and avoid unrealistic nutrient additions. You will see in our experiments we mostly took a smaller portion of the

BGC-Argo trajectories. In the case of Argo 6902547, which we presented in the main text, even though we show the full data in the figures, we avoided taking statistics for the year 2016. The reason is that, we believe that the Argo is entering a different watermass which is unlikely to represent the change in a 1D setup.

---

## Author Response (AR2)

We thank the reviewer for their constructive feedback. We report here our responses to the specific points.

The text in *blue italics* are the comments from the referee. The text in "black regular" is our response to the comment, and the text in *red italics* is our text that appears in the revised manuscript.

Comments from the referee:

*Many of my comments from an earlier revision are resolved, and the only outstanding issue in my mind is the quantification and description of how parameter uncertainty is incorporated into the model, either now or in the future. I saw in your response that you 'are not pursuing a precise 1-to-1 match between the model and BGC-Argo, but exploring notable differences that should be improved regardless of the concentration differences'*

*When looking at the manuscript and performance metrics, it appears as if there is no uncertainty or weighting scheme involved in the calculations (equations 1-4). Incorporation of uncertainty and other parameters is a goal of this framework eventually (although not in this manuscript), I recommend the authors write more specifically and more quantitatively about how uncertainty (specific magnitudes for different variates) can be incorporated into the framework. For example Argo chl, oxygen, bbp have vastly different relative uncertainties ranging from 15% to over 200% in some cases. The tuning meteorological climatologies used also have different uncertainties. A plan for how to tune model parameters would be a nice edit in the new paragraph added (through the last revision) about uncertainty.*

We agree on the necessity of including parameter uncertainty in a study of this kind. The reviewer, however, agrees that it is beyond the context of this study to conduct further experiments on uncertainty. We see this paper as the presentation of the framework, how it is set up, and ways it can be used as a tool for model improvements. We believe we have established this aim here. That being said, while this study does not go into details about parameter fine tuning, that would indeed be the natural application of this framework where uncertainties would play a major role. For this reason, we have expanded the text on uncertainty following the reviewer's recommendations, starting with examples of BGC-Argo uncertainties of various sensors, how we approached this issue in this study (employing ADJUSTED BGC-Argo variables), and how we will approach it in the follow-up study (i.e. increase BGC-Argo tracks, employ multiple BGC-Argo variables for the tuned model parameters to better represent the environmental constraints.) We envision an ensemble suite of model runs, and together with multiple BGC-Argo tracks, we will have a very large dataset to apply the statistical analyses. And as a final step, we will refine the search for optimal parameter sets by narrowing it down for the variables with lower uncertainty and expanding it for those with higher uncertainty.  And as a further follow-up study, a data-assimilation scheme can be included. All this information is added to the Concluding remarks section, and the following is the added/modified text:

[revised manuscript text omitted]